# Connecting genetic risk to disease end points through the human blood plasma proteome

Karsten Suhre[1],**, Matthias Arnold[2],*, Aditya Mukund Bhagwat[3],*, Richard J. Cotton[3],*, Rudolf Engelke[3],*, Johannes Raffler[2],*, Hina Sarwath[3],*, Gaurav Thareja[1],*, Annika Wahl[4,5],*, Robert Kirk DeLisle[6], Larry Gold[6], Marija Pezer[7], Gordan Lauc[7], Mohammed A. El-Din Selim[8], Dennis O. Mook-Kanamori[9], Eman K. Al-Dous[10], Yasmin A. Mohamoud[10], Joel Malek[10], Konstantin Strauch[11,12], Harald Grallert[4,5,13], Annette Peters[5,13,14], Gabi Kastenmüller[2,13], Christian Gieger[4,5,13],** & Johannes Graumann[3],**,†

Genome-wide association studies (GWAS) with intermediate phenotypes, like changes in metabolite and protein levels, provide functional evidence to map disease associations and translate them into clinical applications. However, although hundreds of genetic variants have been associated with complex disorders, the underlying molecular pathways often remain elusive. Associations with intermediate traits are key in establishing functional links between GWAS-identified risk-variants and disease end points. Here we describe a GWAS using a highly multiplexed aptamer-based affinity proteomics platform. We quantify 539 associations between protein levels and gene variants (pQTLs) in a German cohort and replicate over half of them in an Arab and Asian cohort. Fifty-five of the replicated pQTLs are located *in trans*. Our associations overlap with 57 genetic risk loci for 42 unique disease end points. We integrate this information into a genome-proteome network and provide an interactive web-tool for interrogations. Our results provide a basis for novel approaches to pharmaceutical and diagnostic applications.

[1] Department of Physiology and Biophysics, Weill Cornell Medicine-Qatar, Education City, PO 24144 Doha, Qatar. [2] Institute of Bioinformatics and Systems Biology, Helmholtz Zentrum München, German Research Center for Environmental Health, Ingolstädter Landstraße 1, 85764 Neuherberg, Germany. [3] Proteomics Core, Weill Cornell Medicine-Qatar, Education City, PO 24144 Doha, Qatar. [4] Research Unit of Molecular Epidemiology, Helmholtz Zentrum München, German Research Center for Environmental Health, Ingolstädter Landstraße 1, 85764 Neuherberg, Germany. [5] Institute of Epidemiology II, Helmholtz Zentrum München, German Research Center for Environmental Health, Ingolstädter Landstraße 1, 85764 Neuherberg, Germany. [6] SomaLogic, 2945 Wilderness Pl, Boulder, Colorado 80301, USA. [7] Genos Ltd, Glycoscience Research Laboratory, Hondlova 2/11, 10000 Zagreb, Croatia. [8] Department of Dermatology, Hamad Medical Corporation, PO Box 3050 Doha, Qatar. [9] Leiden University Medical Centre, PO Box 9600, 2300 RC Leiden, The Netherlands. [10] Genomics Core, Weill Cornell Medicine-Qatar, Education City, PO 24144, Doha, Qatar. [11] Institute of Genetic Epidemiology, Helmholtz Zentrum München, German Research Center for Environmental Health, Ingolstädter Landstraße 1, 85764 Neuherberg, Germany. [12] Institute of Medical Informatics, Biometry and Epidemiology, Chair of Genetic Epidemiology, Ludwig-Maximilians-Universität, Marchioninistraße 15, 81377 München, Germany. [13] German Center for Diabetes Research (DZD), Ingolstädter Landstraße 1, 85764 Neuherberg, Germany. [14] German Center for Cardiovascular Disease Research (DZHK), Oudenarder Straße 16, 13347 Berlin, Germany. * These authors contributed equally to this work. ** These authors jointly supervised this work. † Present address: Scientific Service Group Biomolecular Mass Spectrometry, Max Planck Institute for Heart and Lung Research, W.G. Kerckhoff Institute, Ludwigstr. 43, D-61231 Bad Nauheim, Germany. Correspondence and requests for materials should be addressed to K.S. (email: karsten@suhre.fr) or to C.G. (email: christian.gieger@helmholtz-muenchen.de) or to J.G. (email: johannes.graumann@mpi-bn.mpg.de).

Genome-wide association studies (GWAS) with intermediate phenotypes, such as metabolite and protein levels, can reveal variations in protein abundances, enzymatic activities, interaction properties or regulatory mechanisms, which can inform clinical applications[1–3]. Studies with aptamer-, immunoassay- and mass-spectrometry-based proteomics (Supplementary Table 1) have shown that protein levels are strongly modulated by variations in the nearby *cis* regions of their encoding genes[4–9]. However, proteomics-based genetic association studies have been limited by small protein panels or cohort sizes, and thus, have not systematically addressed associations across multiple chromosome locations. These *trans*-associations are key in establishing functional links between different risk-variants, because they reveal the downstream effectors in the pathways between *cis*-encoded risk-variants and disease end points.

Here we use a highly multiplexed, aptamer-based, affinity proteomics platform (SOMAscan)[10] in a GWAS to quantify levels of 1,124 proteins in blood plasma samples. We investigate samples from 1,000 individuals of the population-based KORA (Cooperative Health Research in the Region of Augsburg) study[1,11] and replicate the results in 338 participants of the Qatar Metabolomics Study on Diabetes (QMDiab) with participants of Arab and Asian ethnicities[12].

We identify 539 independent, single-nucleotide polymorphism (SNP)-protein associations, which include novel inter-chromosomal (*trans*) links related to autoimmune disorders, Alzheimer's disease, cardiovascular disease, cancer and many other disease end points. Our study represents the outcome of over 1,100 new GWAS with blood circulating protein levels, many of which have never been reported before. We find that up to 60% of the naturally occurring variance in the blood plasma levels of essential proteins can be explained by two or more independent variants of a single gene located on another chromosome, and identify strong genetic associations with intermediate traits in pathways involved in complex disorders.

## Results

**A GWAS with blood circulating protein levels**. We used linear additive genetic regression models, adjusted for relevant covariates, to analyse 509,946 common autosomal SNPs for genome-wide associations with 1,124 protein levels measured in 1,000 blood samples from the KORA study (see Methods). We grouped association signals with correlated SNPs ($r^2 > 0.1$) into single-protein quantitative trait loci (pQTLs) and identified 539 pQTLs (284 unique proteins, 451 independent SNPs) with a conservative Bonferroni level of significance ($P < 8.72 \times 10^{-11} = 0.05/509,946/1,124$; unless otherwise stated we report uncorrected $P$ values for the reported traits from linear regression models). For 21% of the assayed proteins we detected a cis-pQTL at a more liberal significance cutoff of $P < 5 \times 10^{-8}$. The full list is available as Supplementary Data 1. We then fine-mapped our associations with variants imputed from the 1,000-Genomes project database. Next, we attempted replication of 462 associations that had a suitable genotyped tag SNP ($r^2 > 0.8$) in samples from 338 participants of the QMDiab study[12]. Of those, 234 associations (50.6%) were replicated at a Bonferroni level of significance ($P < 1.08 \times 10^{-4} = 0.05/462$), and an additional 150 (totalling to 83.1%) showed nominal significance ($P < 0.05$) in the replication sample. We observed directional consistency between primary and replication sample for 215 of the 234 replicated SNPs. Discrepancy for the remaining 19 SNPs may be explained by changes in major and minor allele coding between the two cohorts and ethnic differences. What is

more, out of 208 associations that had 95% replication power (determined by sampling), 171 (82.2%) were replicated and 198 (95.2%) were nominally significant. Moreover, we replicated several associations previously reported in aptamer-, immunoassay- and mass-spectrometry-based studies, which demonstrated concordance among these technologies (Supplementary Tables 1–5).

**Interconnectedness and annotation of the plasma proteome**. The SOMAscan aptamers were generated to bind specifically to proteins implicated in numerous diseases and physiological processes and to target a broad range of secreted, intracellular and extracellular proteins, which are detectable in blood plasma (Supplementary Fig. 1, Supplementary Data 2). To identify interrelations between these proteins, we used Gaussian graphical modelling (GGM) to connect 1,092 proteins through 3,943 Bonferroni-significant partial protein–protein correlation edges (Supplementary Data 3). Then, we added all 451 genetic pQTL variants as nodes, and connected them to the protein network through the 539 SNP-protein associations (Fig. 1). This network is freely available online, and it can be navigated via an interactive web interface. Overall, we found that given our study power the blood plasma levels of over 20% of all assayed proteins were under substantial genetic control, and in some cases, this control resulted in nearly total protein ablation (Fig. 2). Additionally, a wide-spread feature was the convergence of multiple association signals to impact the levels of key proteins of biomedical and pharmaceutical interest (Supplementary Fig. 2): with the Ingenuity Pathway Analysis database (IPA, Qiagen Inc.), we annotated 50 proteins as clinical or pharmaceutical biomarkers (Supplementary Table 6) and 43 as drug targets (Supplementary Table 7), and all of these had at least one replicated pQTL.

Next, we identified and annotated putative causative and disease-relevant variants. We used the SNiPA web-tool[13] to retrieve variant-specific annotations for all SNPs from the 1,000-Genomes project that were in linkage disequilibrium with an identified pQTL (linkage disequilibrium $r^2 > 0.8$). The annotations included primary effect predictions, SNPs in experimentally identified regulatory elements (ENCODE), expression QTLs (eQTLs) and disease associated variants. Of 384 annotated *cis*-pQTLs, 228 had a variant in the gene coding region, whereof 74 were protein-changing. Eighty-eight had a variant in a regulatory element, and 179 had an eQTL that matched the associated protein. We complemented the pQTL annotation with 122 overlapping methylation QTLs (meQTLs) (Supplementary Data 1) and 14 overlapping metabolic QTLs (mQTLs) (Supplementary Table 8). With the GWAS catalogue as a reference, and by including publicly available summary statistics data from 16 large disease GWAS consortia, we identified 83 GWAS associations for 42 unique disease end points that overlapped with 57 pQTL loci (Supplementary Table 9).

**Trans-pQTLs link genotype to GWAS end points**. *Trans*-associations are exceptionally valuable for identifying new pathways. These associations establish causal links between proteins encoded at the GWAS loci and the blood levels of one or several *trans*-encoded proteins. We identified 148 *trans*-pQTLs and replicated 55. Forty-nine *trans*-pQTLs had 95% replication power, we replicated 38 of these. Six replicated *trans*-pQTLs had an additional replicated *cis*-association, two had two replicated *trans*-associations and one had three replicated *trans*-associations. Three independent SNPs at the *haptoglobin* (*HP*) locus had together four pQTLs, and

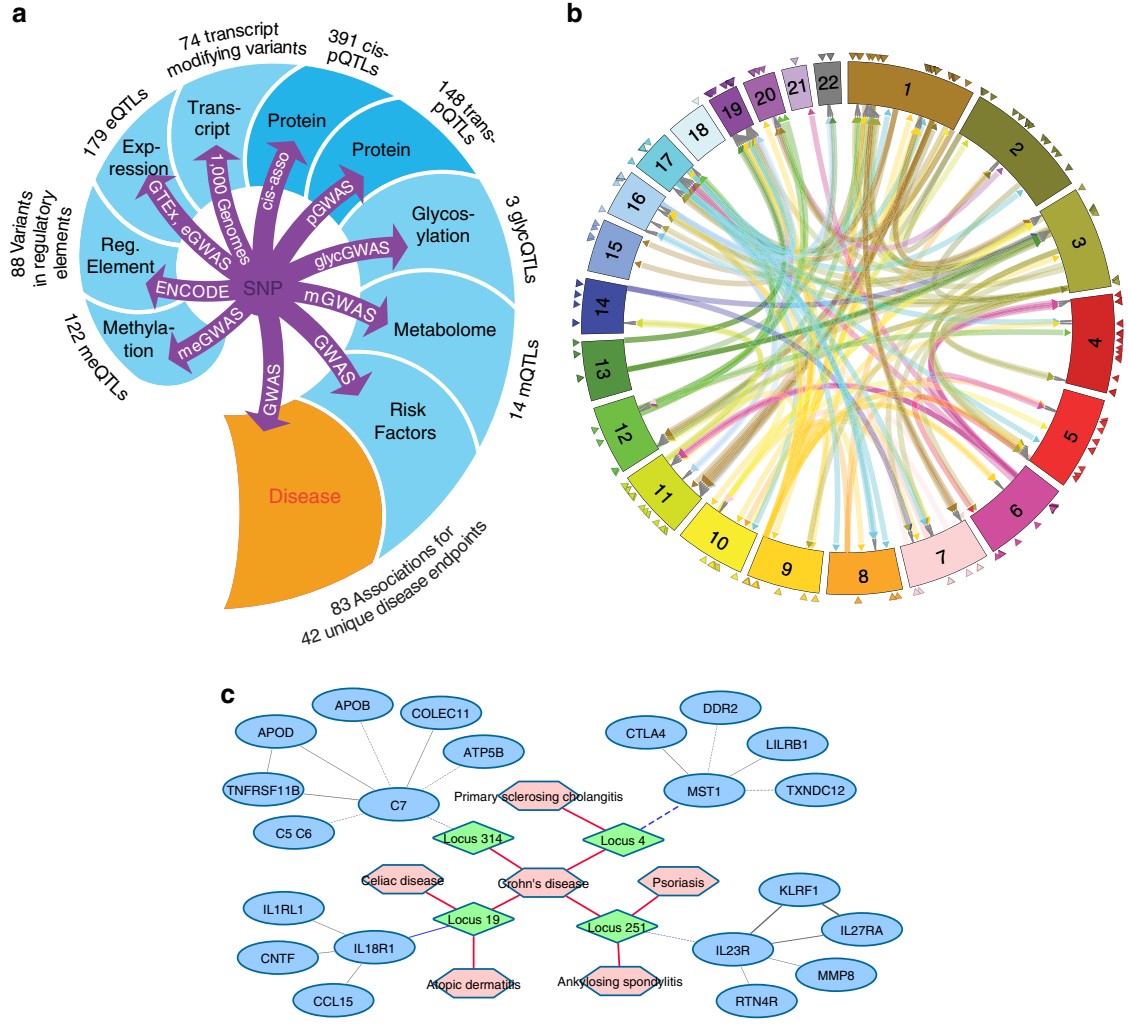

**Figure 1 | The genome-proteome-disease network. (a)** Data sources integrated into the network, indicating the number and type of the overlapping associations, from the SNP to the disease end point; all associations are freely accessible at http://proteomics.gwas.eu. (**b**) Circular plot of all *cis*- and *trans*-associations, *cis*-pQTLs are indicated by triangles, *trans*-pQTLs connect associated variant locations and *trans*-encoded protein locations. An interactive version of this circular plot constitutes an entry point to query the integrated web-server. (**c**) Example of a genome-proteome-disease sub-network obtained from the server for a query using the search word 'Crohn's Disease'. Network elements are disease traits (pink hexagons), pQTL loci (green diamonds), protein levels (blue ovals); nodes are connected by genetic associations, partial correlations and disease GWAS associations. This example (edited here for clarity) revealed four risk loci that associated with plasma levels of C7, MST, IL23R and IL18R, respectively. These four proteins all have a major role in auto-immune disorders. Partial correlations between neighbouring proteins reveal pathways that may be involved in the aetiology of Crohn's disease. Similar networks can be retrieved starting with a query using any of the 539 pQTLs, 1,124 proteins and 42 unique co-associated disease end points. In the integrated web-server, all items are interactively linked to association data from the discovery and the replication study, regional association plots based on imputed variants, locus annotations including co-associated eQTL-, meQTL-, mQTL-, regulatory-, coding- and disease risk-variants, and link-outs to relevant protein databases, original data sources and primary publications. The links in this network reflect the outcome of many natural experiments, represented by genetic variations observed in the genomes of hundreds of individuals from the general population and probed by deep proteomics phenotyping using over 1,100 different aptamers.

two proteins had replicated *trans*-associations at two distinct chromosome locations (Tables 1 and 2).

In this study, the pleiotropic *ABO* blood group gene exhibited the most promiscuous protein association signal. This locus displayed six independent genetic variants that were associated with 14 different proteins through replicated *trans*-pQTLs; three of these associations had been published previously (VWF, SELE, SELP, Supplementary Table 2). The other eleven associations were new, to our knowledge (BCAM, CD200, CD209, CDH5, FLT4, INSR, KDR, MET, NOTCH1, TEK, TIE1).

Genetic variance in *ABO* has been associated with coronary artery disease, stroke[14], and diabetes[15] (Supplementary Fig. 3).

The non-O blood group is one of the most important genetic risk factors for venous thromboembolism[16], pancreatic cancer[17] and susceptibility to infectious diseases[18]. However, we lack a full understanding of the proteins and pathways involved in the pathogenic effects of these genetic variants. Based on the pQTLs reported here, we found support for the following three new hypotheses: First, the association between *ABO* and levels of circulating insulin receptor (INSR) reflects the previously reported associations between the *ABO* locus and diabetes and the insulin receptor and diabetes[15]. The association between *ABO* and the insulin receptor suggests that INSR-mediated insulin signalling may be involved in the ABO-diabetes

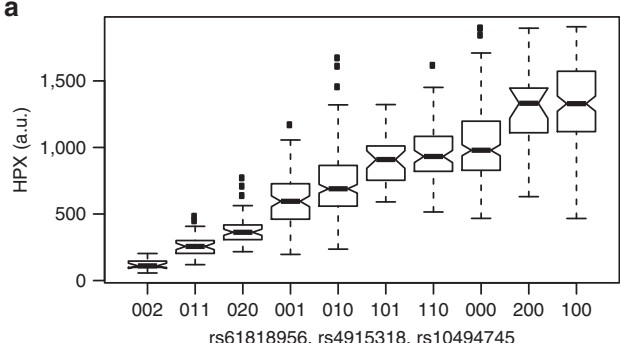

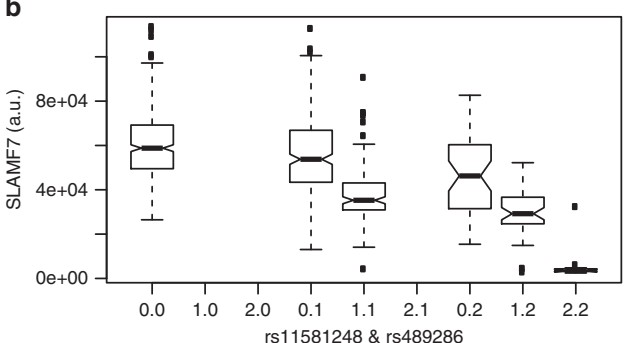

**Figure 2 | Examples of protein levels that are determined by multiple independent genetic variants.** Box-whisker plots of protein levels of (**a**) Haemopexin HPX and (**b**) SLAMF7 as a function of genotype. Data presented from the KORA study ($N = 997$). Whiskers extend to the most extreme data point that still falls within the 1.5 inter-quartile range. The number of minor alleles of the respective genetic variant is given; for instance, in **a**, '002' refers to individuals that are homozygous for the major alleles of rs61818956 and rs4915318 and for the minor allele of rs10494745, and in **b**, '0.2' refers to homozygotes of the major allele of rs11581248 and the minor allele of rs489286. Only variant combinations that were observed in the study population are shown in the case of HPX. SNPs rs61818956, rs4915318 and rs10494745 are located in *trans* in the *CFHR2/CFHR4* gene locus. Further examples are shown in Supplementary Fig. 2.

association. Second, rs651007 associated here with P-selectin (SELP). SELP-positive platelets were previously reported to be associated with blood pressure[19], and rs651007 was associated in a GWAS with angiotensin converting enzyme activity. This variant was further identified as an mQTL for a number of dipeptides in blood[20], which may be produced by the dipeptidase angiotensin converting enzyme[21]. These observations suggest a potential role for the SELP pQTL in the GWAS association between *ABO* and cardiovascular disease. Third, several of the 14 proteins associated here with *ABO* have been shown to interact or form complexes in relation with angiogenesis and vascular maturation processes: The angiopoietin-1 receptor (TEK) has a role in embryonic vascular development and phosphorylates the tyrosine kinase TIE1. TIE1 overexpression in endothelial cells upregulates selectin E (SELE)[22]. Strain induced angiogenesis is mediated in part through a Notch-dependent, Ang1/Tie2 signalling pathway that implicates NOTCH1 and TIE1 (ref. 23). Vascular endothelial (VE)-cadherin (CDH5) is required for normal development of the vasculature in the embryo and for angiogenesis in the adult, and it is associated with VE growth factor (VEGF) receptor-2 (KDR) on the exposure of endothelial cells to VEGF[24]. Heterodimers of KDR and vascular endothelial growth factor receptor 3 (FLT4) positively regulate angiogenic

sprouting[25]. VEGF directly and negatively regulates tumour cell invasion through enhanced recruitment of the protein tyrosine phosphatase 1B (PTP1B) to a hepatocyte growth factor receptor (MET)—KDR hetero-complex[26]. VEGF also synergistically increased tumour necrosis factor-alpha-induced E-selectin messenger RNA (mRNA) and shedding of soluble E-selectin. Synergistic upregulation of E-selectin expression by VEGF is mediated via KDR and calcineurin signalling[27]. These observations suggest that genetic variance in *ABO* has major effects on a network of proteins involved in cell adhesion, angiogenesis and neo-vascularization. Consequently, the here reported *trans*-pQTLs indicate novel pathways that may explain ABO-mediated cancer susceptibility through regulation of vascular metabolism[28].

**Post-translational modifications**. To follow up on the many hypotheses generated by the pQTLs reported here, expert knowledge and experimentation will be required. Given the fact that several of the proteins identified in our pQTLs were glyco-proteins, we performed plasma protein glycoprofiling (see Online Methods) to investigate whether some of our pQTLs were involved in post-translational protein modifications.

For instance, SNP rs3760775, located at the *FUT3* gene locus, was associated with plasma levels of the there encoded galacto-side3(4)-L-fucosyltransferase protein. We found that this same variant was strongly associated with the *N*-glycan GP33 ($P = 1.4 \times 10^{-15}$) (see Fig. 3a for the GP33 glycan structure). In a previous GWAS, SNP rs3760775 was reported to be associated with the glycan antigen, CA19-9 (ref. 29), a widely used cancer biomarker, which is present on multiple proteins[30]. Nearly 20 years ago, studies demonstrated a similar association between CA19-9 and the *FUT3* gene dosage[31,32]. This previously reported association between variance in the *FUT3* locus and the expression of cancer antigen, CA19-9, and our observation of a similar association with GP33, suggest that both glycans may be involved in a same pathway.

Another glycoprofile investigation began with the strong, genotype-dependent correlation between complement factor C4 and the *N*-glycan, GP19 ($P = 2.0 \times 10^{-27}$) (Fig. 3d). We found that SNP rs8283 was associated with both C4 and GP19. GP19 is composed of nine mannose moieties (M9 glycan structure), and it was previously reported to be attached to C4 (ref. 33). Moreover, M9 glycans are the principal target ligand for mannose-binding protein (MBL), which binds to C4 (ref. 34), and subsequently, activates the complement system through the lectin pathway. MBL has been implicated in the pathology of rheumatoid arthritis in many studies, including a recent meta-analysis, which confirmed the association between functional MBL variants and rheumatoid arthritis risk[35]. The rs8283 variant was also associated with rheumatoid arthritis ($P = 3.8 \times 10^{-51}$) (ref. 36), but is not in linkage disequilibrium with the top reported rheumatoid arthritis-risk variants in the HLA region. Hence, SNP rs8283 appears to be an independent signal, possibly mediated through MBL binding to C4 glycans.

**Biomedical relevance**. A major challenge in conducting a disease GWAS is the difficulty in identifying causative variants in the pathophysiological pathways that lead to the observed clinical manifestations. Generally, hypotheses about the identity of the disease-causing genes are based on biological arguments, such as the presence of SNPs in the regulatory or coding regions of functionally plausible genes, which may be supported by co-associated eQTLs. However, a much stronger argument is provided by the co-association with a pQTL, which constitutes firm experimental evidence that the blood levels of

**Table 1 | List of replicated *trans*-pQTLs.**

| Locus | Candidate cis-gene(s) | Sentinel SNP | Chr | Position | Type | *P*value KORA | *P* value QMDiab | Protein name (gene symbol) |
|---|---|---|---|---|---|---|---|---|
| 192 | GJA9[ace], RHBDL2[ace], RP5-864K19.6[ace], RRAGC[ab], MYCBP[ac], … | rs4494114 | 1 | 39,339,682 | Trans | $1.9 \times 10^{-20}$ | $3.0 \times 10^{-8}$ | Kunitz-type protease inhibitor 1 (SPINT1) |
| 210 | C1orf168[bc], C8A[a], C8B[a], DAB1[b] | rs626457 | 1 | 57,407,484 | Trans | $3.3 \times 10^{-19}$ | $1.7 \times 10^{-8}$ | Neurexophilin-1 (NXPH1) |
| 71 | PSRC1[ac], CELSR2[ac], AMPD2[b], SORT1[c] | rs646776 | 1 | 109,818,530 | Trans | $1.0 \times 10^{-52}$ | $1.7 \times 10^{-18}$ | Granulins (GRN) |
| 413 | F5[ae] | rs9332653 | 1 | 169,490,772 | Cis | $1.6 \times 10^{-11}$ | $7.5 \times 10^{-5}$ | Coagulation Factor V (F5) |
| | | | | | Trans | $1.9 \times 10^{-11}$ | $1.5 \times 10^{-5}$ | Calcium/calmodulin-dependent protein kinase type 1 (CAMK1) |
| 17 | F5[ae], SELP[b], SELL[b] | rs4525 | 1 | 169,511,734 | Trans | $2.0 \times 10^{-110}$ | $1.3 \times 10^{-32}$ | Calcium/calmodulin-dependent protein kinase type 1 (CAMK1) |
| 98 | CFH[a], CFHR3[c] | rs6695321 | 1 | 196,675,861 | Trans | $9.4 \times 10\text{-}40$ | $5.1 \times 10^{-5}$ | Complement C1s subcomponent (C1S) |
| 72 | CFHR4[ae], CFHR2[a], ASPM[a], ZBTB41[a] | rs10494745 | 1 | 196,887,457 | Trans | $1.8 \times 10^{-52}$ | $2.4 \times 10^{-11}$ | Haemopexin (HPX) |
| 76 | CFHR4[ac], CFHR2/5[a],CFHR1/3[c], CFH[c] | rs10801582 | 1 | 196,944,357 | Trans | $1.1 \times 10^{-49}$ | $2.0 \times 10^{-11}$ | Haemopexin (HPX) |
| 122 | TRIM58[ae] | rs1339847 | 1 | 248,039,294 | Trans | $9.2 \times 10^{-33}$ | $5.4 \times 10^{-6}$ | Dynein light chain roadblock-type 1 (DYNLRB1) |
| 93 | COLEC11[ac], ALLC[c] | rs7588285 | 2 | 3,648,186 | Cis | $1.1 \times 10^{-40}$ | $1.4 \times 10^{-16}$ | Collectin-11 (COLEC11) |
| | | | | | Trans | $2.7 \times 10^{-37}$ | $8.7 \times 10^{-20}$ | Interleukin-19 (IL19) |
| 157 | LTF[abe], CCR5[b], CCR3[c] | rs1126478 | 3 | 46,501,213 | Trans | $5.1 \times 10^{-26}$ | $4.1 \times 10^{-13}$ | Alkaline phosphatase, tissue-nonspecific isozyme (ALPL) |
| 95 | IP6K2[abce], CELSR3[abc], NCKIPSD[abc], ARIH2[abc], USP19[abe],… | rs11715835 | 3 | 48,770,732 | Trans | $2.2 \times 10^{-40}$ | $4.2 \times 10^{-8}$ | Thioredoxin domain-containing protein 12 (TXNDC12) |
| 91 | RBM6[ac], RNF123[ae], BSN[a], AMIGO3[a], GMPPB[a],… | rs4688759 | 3 | 50,008,118 | Trans | $1.8 \times 10^{-42}$ | $4.2 \times 10^{-8}$ | Thioredoxin domain-containing protein 12 (TXNDC12) |
| 52 | DCBLD2[c], CPOX[c], ST3GAL6[c] | rs10935480 | 3 | 98,431,986 | Trans | $9.9 \times 10^{-70}$ | $1.2 \times 10^{-17}$ | Vascular endothelial growth factor receptor 3 (FLT4) |
| 352 | DNAJC13[abc], ACAD11[abe], NPHP3[ac], ACKR4[a], UBA5[a],… | rs17412738 | 3 | 132,257,419 | Trans | $5.3 \times 10^{-13}$ | $3.8 \times 10^{-5}$ | C-C motif chemokine 21 (CCL21) |
| 203 | PCOLCE2[ac], U2SURP[b], ATR[b], PLS1[b] | rs4683702 | 3 | 142,617,138 | Trans | $2.0 \times 10^{-19}$ | $8.3 \times 10^{-5}$ | Endothelin-converting enzyme 1 (ECE1) |
| 31 | HRG[ae] | rs2228243 | 3 | 186,395,113 | Cis | $4.7 \times 10^{-94}$ | $2.7 \times 10^{-25}$ | Histidine-rich glycoprotein (HRG) |
| | | | | | Trans | $4.2 \times 10^{-82}$ | $6.0 \times 10^{-18}$ | Dual specificity mitogen-activated protein kinase kinase 4 (MAP2K4) |
| 43 | HRG[ae] | rs1042445 | 3 | 186,395,436 | Trans | $8.4 \times 10^{-78}$ | $1.2 \times 10^{-22}$ | Dual specificity mitogen-activated protein kinase kinase 4 (MAP2K4) |
| 26 | KNG1[ae] | rs2304456 | 3 | 186,445,052 | Cis | $2.9 \times 10^{-97}$ | $2.8 \times 10^{-40}$ | Kininogen-1 (KNG1) |
| | | | | | Trans | $1.4 \times 10^{-10}$ | $1.5 \times 10^{-6}$ | Leucine carboxyl methyltransferase 1 (LCMT1) |
| 96 | KNG1[ae] | rs5030062 | 3 | 186,454,180 | Trans | $3.0 \times 10^{-40}$ | $1.7 \times 10^{-13}$ | Coagulation Factor XI (F11) |
| | | | | | Trans | $1.8 \times 10^{-35}$ | $8.0 \times 10^{-10}$ | Plasma kallikrein (KLKB1) |
| 123 | SKIV2L[ac], C2[a], NELFE[a], DXO[a], STK19[a],… | rs9283893 | 6 | 31,897,219 | Trans | $1.2 \times 10^{-32}$ | $1.6 \times 10^{-28}$ | Neutrophil collagenase (MMP8) |
| 159 | SKIV2L[ace], C4B[ac], TNXB[ab], DXO[a], STK19[a],… | rs387608 | 6 | 31,941,557 | Trans | $2.5 \times 10^{-25}$ | $2.4 \times 10^{-8}$ | gp41 C34 peptide, HIV (Human-virus) |
| 182 | TAP2[a], PSMB9[a], TAP1[a], PSMB8[a], COL11A2[b],… | rs17220241 | 6 | 32,822,244 | Trans | $9.6 \times 10^{-22}$ | $1.3 \times 10^{-5}$ | Alpha-2-macroglobulin receptor-associated protein (LRPAP1) |
| 417 | ZFPM2[a], CXCL5[d] | rs16873418 | 8 | 106,592,145 | Trans | $1.9 \times 10^{-11}$ | $1.5 \times 10^{-5}$ | Tumour necrosis factor receptor superfamily member EDAR (EDAR) |
| 442 | ABO[ac], SURF1[c], SLC2A6[c], GBGT1[c] | rs7857390 | 9 | 136,128,546 | Trans | $5.8 \times 10^{-11}$ | $1.0 \times 10^{-6}$ | Tyrosine-protein kinase receptor Tie-1, soluble (TIE1) |
| 69 | ABO[abce], OBP2B[bc], DBH[b], SURF1/2[b], ADAMTSL2[b],… | rs8176749 | 9 | 136,131,188 | Trans | $6.1 \times 10^{-53}$ | $2.7 \times 10^{-42}$ | Cadherin-5 (CDH5) |
| | | | | | Trans | $1.7 \times 10^{-51}$ | $3.5 \times 10^{-27}$ | Tyrosine-protein kinase receptor Tie-1, soluble (TIE1) |
| | | | | | Trans | $1.1 \times 10^{-35}$ | $2.0 \times 10^{-10}$ | Angiopoietin-1 receptor, soluble (TEK) |
| | | | | | Trans | $1.5 \times 10^{-11}$ | $5.8 \times 10^{-5}$ | Basal cell adhesion molecule (BCAM) |
| | | | | | Cis* | $6.0 \times 10^{-10}$ | $5.3 \times 10^{-6}$ | Neurogenic locus notch homolog protein 1 (NOTCH1) |
| 354 | ABO[ac], SURF6[c] | rs8176720 | 9 | 136,132,873 | Trans | $7.4 \times 10^{-10}$ | $6.8 \times 10^{-8}$ | Insulin receptor (INSR) |

Please see the legend of Table 2 for a description of the replicated *trans*-pQTL loci.
*NOTCH1 is encoded in cis on chromosome 9, but distant from ABO.

**Table 2 | List of replicated *trans*-pQTLs**

| Locus | Candidate cis-gene(s) | Sentinel SNP | Chr | Position | Type | P value KORA | P value QMDiab | Protein name (gene symbol) |
|---|---|---|---|---|---|---|---|---|
| 36 | ABO[abc], TSC1[b], AK8[b], SARDH[b], GBGT1[c] | rs505922 | 9 | 136,149,229 | Trans | $1.2 \times 10^{-20}$ | $1.0 \times 10^{-8}$ | von Willebrand factor (VWF) |
| | | | | | Trans | $7.6 \times 10^{-86}$ | $4.1 \times 10^{-28}$ | CD209 antigen (CD209) |
| 269 | ABO[ac], DBH[b], ADAMTSL2[b], SARDH[b], RALGDS[b],... | rs630510 | 9 | 136,149,581 | Trans | $1.6 \times 10^{-15}$ | $2.4 \times 10^{-10}$ | Tyrosine-protein kinase receptor Tie-1, soluble (TIE1) |
| 28 | ABO[ac], DBH[b], ADAMTSL2[b], SARDH[b], RALGDS[b],... | rs651007 | 9 | 136,153,875 | Trans | $1.2 \times 10^{-96}$ | $8.2 \times 10^{-23}$ | E-Selectin (SELE) |
| | | | | | Trans | $3.9 \times 10^{-44}$ | $5.2 \times 10^{-15}$ | Insulin receptor (INSR) |
| | | | | | Trans | $1.1 \times 10^{-31}$ | $1.4 \times 10^{-8}$ | Vascular endothelial growth factor receptor 3 (FLT4) |
| | | | | | Trans | $3.3 \times 10^{-19}$ | $1.2 \times 10^{-9}$ | Hepatocyte growth factor receptor (MET) |
| | | | | | Trans | $3.1 \times 10^{-13}$ | $1.9 \times 10^{-5}$ | Vascular endothelial growth factor receptor 2 (KDR) |
| | | | | | Trans | $7.4 \times 10^{-13}$ | $3.4 \times 10^{-6}$ | P-Selectin (SELP) |
| | | | | | Trans | $8.0 \times 10^{-11}$ | $8.2 \times 10^{-6}$ | OX-2 membrane glycoprotein (CD200) |
| 79 | CPN1[a], HIF1AN[c] | rs7091871 | 10 | 101,810,304 | Trans | $1.0 \times 10^{-48}$ | $3.5 \times 10^{-16}$ | Calcium/calmodulin-dependent protein kinase type 1 (CAMK1) |
| | | | | | Trans | $1.4 \times 10^{-12}$ | $1.7 \times 10^{-7}$ | Calpastatin (CAST) |
| 150 | SIK3[ab], SIDT2[bc], PCSK7[b], BUD13[b], RNF214[b],... | rs12099358 | 11 | 116,726,048 | Trans | $1.6 \times 10^{-26}$ | $1.9 \times 10^{-8}$ | Beta-endorphin (POMC) |
| 65 | OAF[abe], POU2F3[bc], ARHGEF12[b], TMEM136[b], TRIM29[b],... | rs692804 | 11 | 120,099,368 | Trans | $1.1 \times 10^{-56}$ | $5.3 \times 10^{-24}$ | Interleukin-25 (IL25) |
| 115 | C1S[abce], C1RL[c] | rs12146727 | 12 | 7,170,336 | Cis | $3.4 \times 10^{-35}$ | $3.6 \times 10^{-7}$ | Complement C1r subcomponent (C1R) |
| | | | | | Trans | $2.0 \times 10^{-15}$ | $8.8 \times 10^{-6}$ | Complement C1q subcomponent (C1QA C1QB C1QC) |
| 304 | POC1B-GALNT4[ace], GALNT4[ace], POC1B[ac], ATP2B1[c] | rs722414 | 12 | 89,937,437 | Trans | $2.3 \times 10^{-14}$ | $3.1 \times 10^{-6}$ | CMRF35-like molecule 6 (CD300C) |
| 8 | ZC3H13[ae], CPB2[ae], LCP1[c] | rs1926447 | 13 | 46,629,944 | Trans | $2.2 \times 10^{-145}$ | $4.3 \times 10^{-5}$ | MAP kinase-activated protein kinase 3 (MAPKAPK3) |
| 155 | PROZ[ac], PCID2[a], CUL4A[a] | rs515863 | 13 | 113,839,747 | Trans | $2.5 \times 10^{-26}$ | $6.2 \times 10^{-9}$ | Dual specificity mitogen-activated protein kinase kinase 2 (MAP2K2) |
| 67 | DHX38[ac], TXNL4B[a], PMFBP1[a], HPR[c], DHODH[c], HP[c] | rs9302635 | 16 | 72,144,174 | Cis | $6.3 \times 10^{-54}$ | $6.2 \times 10^{-19}$ | Haptoglobin (HP) |
| | | | | | Trans | $2.8 \times 10^{-10}$ | $1.4 \times 10^{-8}$ | Ferritin (FTH1 FTL) |
| 82 | SARM1[ac], VTN[a], SLC46A1[a], TMEM199[c], POLDIP2[c], TMEM97[c] | rs2239908 | 17 | 26,725,265 | Trans | $9.5 \times 10^{-48}$ | $3.1 \times 10^{-10}$ | Semaphorin-3 A (SEMA3A) |
| | | | | | Trans | $3.0 \times 10^{-24}$ | $3.9 \times 10^{-8}$ | Calcium/calmodulin-dependent protein kinase type 1D (CAMK1D) |
| | | | | | Trans | $1.1 \times 10^{-10}$ | $7.7 \times 10^{-5}$ | WNT1-inducible-signalling pathway protein 1 (WISP1) |
| 54 | APOC4[ae], APOC4-APOC2[ae], APOC2[a], APOE[c], APOC1[c],... | rs5167 | 19 | 45,448,465 | Trans | $2.1 \times 10^{-69}$ | $9.5 \times 10^{-6}$ | Granulocyte colony-stimulating factor (CSF3) |
| 106 | TRPC4AP[abc], EDEM2[abc], PROCR[ace], GSS[ab], MYH7B[ab],... | rs867186 | 20 | 33,764,554 | Trans | $5.5 \times 10^{-38}$ | $9.0 \times 10^{-24}$ | Vitamin K-dependent protein C (PROC) |

Loci that comprise at least one replicated *trans*-association. Loci are referenced in this study by numbers ranging from 1 to 451 (strongest to weakest) and sorted here by chromosome position. P values are for the association with inverse-normal scaled protein levels from linear regression with genotype; see Supplementary Data 1 for full data of all 539 SNP-protein associations at 451 loci, including statistics for association with alternatively raw- and log-normal-scaled protein levels and estimated replication power. Candidate genes for the protein associations were annotated by considering the following criteria: a variant in linkage disequilibrium with the sentinel SNP (r2 > 0.8) is located in the gene transcript (superscript a), a variant hits a regulatory element of that gene (superscript b), a variant is a *cis*-eQTL (superscript c), a variant is a *trans*-eQTL (superscript d), a variant is protein changing (superscript e). The list of candidate genes in this table is limited to the five most plausible candidate genes for each locus. The full list is available online and in Supplementary Data 1. Every *trans*-pQTL implies the existence of a functional and causal link between a *cis*-encoded candidate gene and the *trans*-encoded target protein(s).

disease-associated proteins vary in response to changes in the genome. Moreover, partial correlations to functionally related proteins, as reported here, may further substantiate and extend hypotheses generated from pQTLs (see example in Fig. 1c). In the following sections, we show how pQTLs identified in this study reveal new insights into multiple disease-associated pathways identified in previous GWAS.

**Auto-immune disorders**. Ankylosing spondylitis is a common cause of inflammatory arthritis and affects one in 200 Europeans.

Evans *et al.*[37] identified two independent ankylosing spondylitis-risk variants in the *endoplasmic reticulum aminopeptidase 1 (ERAP1)* gene; they reported that the major allele of rs30187 and the minor allele of rs10050860 were protective. ERAP1 is involved in trimming peptides before HLA class I presentation. It has recently attracted attention as a drug target for auto-immune disorders[38]. Several studies showed that ERAP1 was present in blood, localized to exosome-like vesicles and present in the extracellular space[39]. Here we found that the two previously identified ankylosing spondylitis-risk variants were associated,

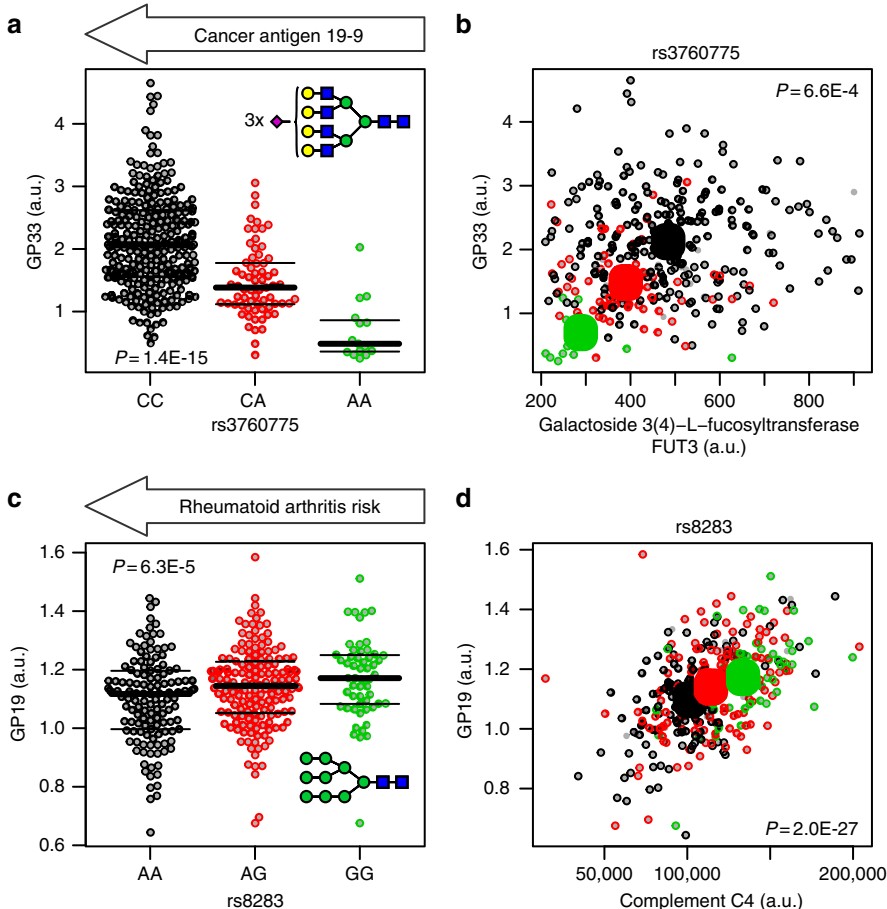

**Figure 3 | Genotype-dependent co-associations of the plasma proteome and the plasma *N*-glycome.** Bee swarm plots of total plasma *N*-glycans GP19 and GP33 (% of total *N*-glycan content) as a function of rs3760775 and rs8283 genotype, respectively, see inset for glycan structure, Blue squares: *N*-acetylglucosamine, green circles: mannose, yellow circles: galactose, purple diamonds: *N*-acetylneuraminic acid; Scatter plots of total plasma *N*-glycans GP19 and GP33 as a function of *Complement factor 4 (C4)* and *Galactoside 3(4)-L-fucosyltransferase (FUT3)* genotype (raw data), respectively (**b**,**d**), black: major allele homozygotes, red: heterozygotes, green: minor allele homozygotes. Large circles indicate means by genotype. *P* values are for the association of glycans with genotype (**a**,**c**) and of glycans with protein levels (**b**,**d**). *P* values are uncorrected from linear regression. The major allele variant of SNP rs3760775 was reported to be associated with the cancer antigen 19-9 and that of SNP rs8283 with increased risk of rheumatoid arthritis.

in an additive manner, with increasing levels of circulating ERAP1 protein (Fig. 4a). Similarly, mRNA sequences isolated from lymphoblastoid cells showed that ERAP1 mRNA expression also increased with increasing numbers of ankylosing spondylitis-risk alleles (Fig. 4b). Previous work concluded that the association between *ERAP1* and ankylosing spondylitis was mainly driven by genetic differences in how ERAP1 enzymatic activity shaped the HLA-B27 peptidome[40,41]. However, our findings suggest that the auto-immunogenic effects of different ERAP1 protein variants may in addition be modulated by genotype-dependent regulation of protein expression. This observation may have broader implications for treatment approaches to autoimmune disorders that depend on ERAP1 antigen processing.

**Complement system and haem clearance.** We identified two independent variants (rs10494745 and rs10801582), located in the *complement factor H-related 2/4 (CFHR2/CFHR4)* gene locus. These variants were associated in *trans* with haemopexin (HPX) protein levels (Fig. 2a). Lower CFHR4 expression levels were associated with lower HPX protein levels. HPX binds haem with high affinity and transports it from the plasma to the liver, which

prevents the accumulation of oxidative species. The rs10494745 variant is a G→E amino acid substitution in CFHR4. It is an eQTL for *CFHR4* expression in liver, as it tags the GTEx rs4915318 variant (GTEx, $P = 1.6 \times 10^{-7}$). Imputed data revealed a third, strong, and independent signal on SNP rs61818956 ($P = 1.13 \times 10^{-74}$), which is located in an intron in the *CFHR2/CFHR4* locus. Conditional analysis showed that all three variants were statistically independent, and together, they explained a surprising 61% of the observed variance in a key protein responsible for oxidative stress reduction. Previous studies reported that CFHR4 interacts with complement component 3 (C3) (ref. 42), and in turn, C3 interacts with haem[43]. Those findings suggest a plausible *trans*-acting pathway that links CFHR4 and HPX. That observation may have important consequences on our understanding of the pathologies involving the classical and alternative complement activation pathways.

**Alzheimer's disease (AD) and mRNA splicing.** Our findings on the major AD-risk variant, rs4420638, may generate particular medical interest. This variant displayed a *cis*-association with increased levels of apolipoprotein E (isoform E2) (APOE) and a concordant *trans*-association with decreased levels of small

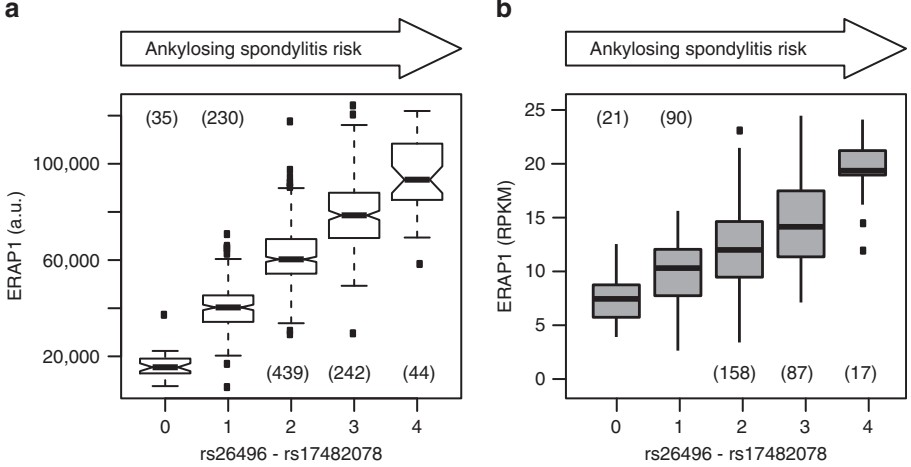

**Figure 4 | Protein and mRNA expression levels of endoplasmic reticulum aminopeptidase 1 (ERAP1) as a function of two ankylosing spondylitis (AS)-risk alleles.** Box-whisker plots of (**a**) ERAP1 blood circulating protein levels and (**b**) *ERAP1* mRNA expression levels observed in lymphoblastoid cells as a function of the sum of AS-risk alleles (minor allele of rs26496, $r^2 = 0.46$ with rs30187; major allele of rs17482078, $r^2 = 0.96$ with rs10050860); the number of individuals per genotype combination is in parentheses; whiskers extend to the most extreme data point that still falls within the 1.5 inter-quartile range.

nuclear ribonucleoprotein F (SNRPF aka RUXF). This dual association was further supported by the finding that the ratio between APOE and SNRPF strengthened the association with rs4420638 by 16 orders of magnitude (p-gain statistic[44]). Although this association lacked sufficient replication power, both APOE and SNRPF associations remained nominally significant in the QMDiab cohort ($P < 0.012$), with concordant trends. SNRPF is a core component of U1, U2, U4 and U5 small nuclear ribonucleoproteins, which are the building blocks of the spliceosome. Recently, a knock-down of U1-70 K or inhibition of U1 small nuclear ribonucleoprotein components was shown to increase the levels of amyloid precursor protein[45]. An association between this major AD-risk variant and a protein of the spliceosomal machinery has not been reported previously. Taken together, these observations support the implication that protein splicing may be an important factor in AD. Furthermore, the *trans*-association between SNP rs4420638 and SNRPF had opposite directionality compared with that with APOE levels. These observations suggest that a regulatory mediator is most likely involved. Theoretically, pharmacological targeting of this mediator could cause an increase in SNRPF, which would increase splicing, and potentially decrease amyloid precursor protein levels.

**Pharmacogenetics.** A pQTL that implicates a drug target may affect patient response to treatment. For example, we observed a strong, replicated *cis*-association between rs489286 and reduced blood levels of the signalling lymphocytic activation molecule-F7 (SLAMF7) in carriers of the minor allele. This was further confirmed with a *SLAMF7*-eQTL that showed identical directionality in mRNA sequencing data from lymphoblastoid cell lines (Supplementary Fig. 4). Furthermore, imputed data identified a strong association between SLAMF7 protein levels and a SNP in the *SLAMF7* intron, rs11581248; heterozygous alleles caused severely reduced SLAMF7 levels, and the homozygous minor allele nearly ablated the SLAMF7 protein (Fig. 2). SLAMF7 is targeted by the recently FDA-approved cancer drug, Elotuzumab, a humanized monoclonal antibody prescribed for relapsed or refractory multiple myeloma. Previously, a small study on Japanese women indicated an association between rs17313034 ($r^2 = 0.82$ with rs489286) and cervical cancer[46]; it showed that the minor allele had a protective

effect. Taken together, these data suggest the possibility that the response to Elotuzumab treatment may depend on the patient's rs489286 genotype, and that homozygotes of the rarer rs11581248 variant (1.5% of the population) may not respond to Elotuzumab. We cannot exclude the possibility that rs17313034 affects the SLAMF7 epitope, which could potentially alter SLAMF7-aptamer binding. However, such an alteration might then also affect Elotuzumab binding. This hypothesis can be tested retrospectively in the phase-3 clinical trial cohort[47].

**Drug target validation.** Plenge *et al.*[48] suggested that naturally occurring genetic variance could be used to validate drug targets, based on genotype–phenotype dose–response curves. For example, IL6R was previously proposed as a target for preventing coronary heart disease. Recently, tocilizumab, a humanized antibody that targets IL6R, was developed and is currently approved for treating rheumatoid arthritis. Plenge *et al.*[48] required that, for validating drug targets, the gene must include multiple causative variants of known biological function. A large IL6R Mendelian randomization analysis[49] found that SNP rs7529229 was associated with increased IL6R, reduced CRP, reduced fibrinogen, and reduced odds of coronary heart disease ($P = 1.53 \times 10^{-5}$). The CardiogramPlus consortium GWAS data confirmed that association ($P = 1.66 \times 10^{-8}$). In the present study, we replicated the IL6R-pQTL (rs4129267, $r^2 = 0.94$ with rs7529229). Furthermore, we identified a second SNP in IL6R, rs11804305, which tags an independent and causative variant, since rs4129267 was already established as functional. Therefore, these two SNPs could be used to investigate dose–response curves in the huge data set available from the IL6R-Mendelian randomization consortium[49]. A similar approach is now feasible for all other protein drug targets that were associated here with multiple pQTLs (for examples see Supplementary Fig. 2). What is more, the aptamers that were used in this study to target these proteins can readily be used as intermediate readouts in assessing drug responses and in optimizing the efficacy of lead components.

**Application to disease GWAS.** Genetic associations with intermediate traits are generally much stronger than associations with disease end points, due to their proximity to the causative variant,

as shown in our previous GWAS with metabolic traits[50]. Therefore, pQTLs can serve as proxies to fine-map genetic disease associations. This approach can be used to identify potentially causative genes and additional independent genetic signals at a locus identified in a disease-GWAS. In particular, pQTLs can be used to identify true positive associations among associations that do not reach genome-wide significance. For instance, rs12146727 was associated with cardiovascular disease (CVD; $P = 6.18 \times 10^{-5}$) in Cardio-gramPlus and with AD ($P = 3.10 \times 10^{-5}$ in the discovery cohort (st1), and $P = 8.27 \times 10^{-6}$ in a combined analysis of a larger cohort (st12comb)) in the International Genomics of Alzheimer's Project. A true positive association requires that the association signal must be strengthened with increasing sample numbers, which was the case for the st1 and st12comb cohorts. In the present study, rs12146727 was replicated as a cis-pQTL for complement C1r subcomponent (C1R), and it was replicated as a trans-pQTL for complement C1q subcomponent (C1QA/C1QB/C1QC). These associations support growing evidence that suggests that the complement system has a role in both, AD and CVD[51]. This example also shows how variants that associate with a disease end point could generate new hypotheses about the role of the co-associated proteins in the disease aetiology. Note that even when these variants have small effect sizes or odds ratios, the related pathways may show large responses to pharmaceutical alteration of these proteins. Hypotheses generated with this approach can be directly tested in animal models, and the existing aptamers can be potentially used as intermediate functional readouts in the drug development process.

## Discussion

Genetic studies with yeast[52] and lymphoblastoid cell lines[53–55] have indicated that cellular protein levels are under strong genetic control. This control was confirmed by the discovery of many cis-acting genetic variants in the human blood proteome[4–9]. Here we described the first large-scale proteomics GWAS on blood plasma proteins derived from a human population. This GWAS represented over 1.1 million individual aptamer binding experiments. By design, our panel of more than 1,100 aptamer targets was highly enriched in biomedically relevant blood circulating proteins, which was reflected in the large overlap of pQTLs with risk loci identified in disease-GWAS. The data generated from this study can be used in future investigations to identify and validate causative variants identified in disease GWAS. For instance, aptamer-based read-outs can be used as intermediate traits in CrispR-based experiments, as exemplified with the FTO-obesity association described by Claussnitzer et al.[56,57].

Surprisingly, we found that up to 60% of the naturally occurring variance in the blood plasma levels of essential proteins could be explained by two or more independent variants of a single gene located on another chromosome (trans-associations). We identified strong genetic associations with intermediate, and most likely functional, traits related to proteins involved in complex disorders. While many of our associations connect these proteins to disease pathways through shared associations, it should be borne in mind that confounding is always a possibility and needs to be ruled out by further experimentation. Our data can be used in future studies to establish dose-response curves for drug-target validation[48]. A greater understanding of the genetic control of circulating levels of protein drug targets and biomarkers may improve pharmaceutical interventions and clinical trials. Because all the aptamers used in this study were synthetically generated and

are well defined, they can be readily developed into specific assays for precise clinical applications. For instance, the SLAMF7-binding aptamer identified here, which revealed a potential genetic effect on Elotuzumab, may be developed directly into a clinical assay for identifying differential responders to immunotherapy. Moreover, this concept can be generalized to other drug targets and biomarkers.

Despite the variable baseline conditions among the participants of the replication cohort (not fasting, high prevalence of diabetes and multiple ethnic backgrounds), we replicated 82% of all sufficiently powered associations. This result emphasized the robustness of the replicated associations. It also suggests that many of the Bonferroni-significant associations that could not be replicated in our QMDiab cohort may be replicated in future studies. About 20% of all assayed proteins had a significant pQTL association. This number is expected to increase in future more highly powered studies.

Genetic variance in a protein sequence may affect its higher order structure, and thus, its aptamer-binding affinity. Similarly, alterations in protein structure may affect binding and specificity in immunoassay-based methods and protein mass in targeted mass spectrometry (MS)-based methods. These issues remain to be addressed. In this study, we showed that structural-based epitope effects on a particular pQTL may be identified or ruled out in several ways, including: allele-specific transcription analysis (for example, CPNE1, Supplementary Fig. 5), co-associated eQTLs, particularly when they include multiple genetic variants (for example, ERAP1, Fig. 4), the absence of correlated SNPs that alter the protein structure (for example, based on SNiPA-annotated 1000 Genomes data), and replication on different platforms (Supplementary Tables 2, 4 and 5). We did not directly follow up on our results using mass spectroscopy. However, Ngo et al.[58] recently showed that response curves for selected aptamer-enriched proteins were linear over a wide dynamic range of spiked protein concentrations. Most importantly, trans-associations, which were a central focus of this study, are not affected by this type of potential artefact.

In summary, our GWAS demonstrated the power of linking the genome to disease end points via the blood proteome. As we have shown in the examples, mining of our data can reveal a plethora of new insights into biological processes and provide a wealth of functional information that is beyond the focus of a single publication. Therefore, we have provided additional interpretations of selected loci in Supplementary Note 1. Furthermore, our complete results are freely accessible for further analyses of pQTL associations with past and future disease GWAS, on an integrated web-server at http://proteomics.gwas.eu.

## Methods

**Study population.** The KORA F4 study is a population-based cohort of 3,080 subjects living in southern Germany. Study participants were recruited between 2006 and 2008 comprising individuals who, at that time, were aged 32–81 years. KORA F4 is the follow-up study of the KORA S4 survey conducted in 1999–2001 (4,261 participants). The study design, standardized sampling methods and data collection (medical history, questionnaires, anthropometric measurements) have been described in detail elsewhere (ref. 11 and references therein). For this study, 1,000 individuals were randomly selected from a subset of 1,800 already deeply phenotyped KORA F4 study participants[1,59,60]. All study participants have given written informed consent and the study was approved by the Ethics Committee of the Bavarian Medical Association.

The QMDiab is a cross-sectional case-control study that was conducted between February and June 2012 at the Dermatology Department of HMC in Doha, Qatar. QMDiab has been described previously and comprises male and female participants in near equal proportions, aged between 23 and 71 years, mainly from Arab, South Asian and Filipino descent[12]. The initial study was approved by the Institutional Review Boards of HMC and Weill Cornell Medicine—Qatar (WCM-Q) (research protocol #11131/11). Written informed consent was obtained from all participants.

**Blood sampling.** Blood samples for omics-analyses and DNA extraction were collected between 2006 and 2008 as part of the KORA F4 follow-up. To avoid variation due to circadian rhythm, blood was drawn in the morning between 08:00 and 10:30 after a period of at least 10 h overnight fasting. Blood was collected without stasis and was kept at 4 °C until centrifugation. The material was then centrifuged for 10 min (2,750$g$ at 15 °C). Plasma samples were aliquoted and stored at −80 °C until assayed on the SOMAscan platform. For QMDiab, non-fasting plasma specimens were collected in the afternoon, after the general operating hours of the morning clinic, and processed using standardized protocols. Cases and controls were collected as they became available, in a random pattern and at the same location using identical protocols, instruments and study personnel. Samples from cases and controls were processed in the lab blinded to their identity. After collection, the samples were stored on ice for transportation to WCM-Q. Within six hours after sample collection all samples were centrifuged at 2,500$g$ for 10 min, aliquoted, and stored at -80 °C until analysis.

**Genotyping.** The Affymetrix Axiom Array was used to genotype 3,788 participants of the KORA S4 study. After thorough quality control (total genotyping rate in the remaining SNPs was 99.8%) and filtering for minor allele frequency >1%, a total of 509,946 autosomal SNPs was kept for the GWAS analysis. One-thousand Genomes Project-imputed genotypes were used for fine mapping and generation of regional association plots (Supplementary Fig. 6). KORA genotype data has been reported previously with many GWAS and was used as-provided by the consortium. We, therefore, do not repeat details here[11].

DNA was extracted from 359 samples from QMDiab and genotyped by the WCM-Q genomics core facility using the Illumina Omni 2.5 array (version 8). High-quality genotype data of 2,338,671 variants was obtained for 353 samples; data for six samples was excluded due to a low overall call rate (<90%). Removal of duplicate variants left 2,327,362 variants. In all, 134,830 variants were removed due to missing genotype data (PLINK option --geno 0.02), leaving 2,192,532 variants. In all, 941,058 variants were removed due to minor allele threshold (PLINK option --maf 0.05), leaving 1,251,474 variants. In all, 28,175 variants were removed due to violation of Hardy-Weinberg equilibrium (PLINK option --hwe 1E-6), leaving 1,223,299 variants. Of these variants, 1,221,345 were autosomal variants. The total genotyping rate of these remaining variants was 99.7%.

**Proteomics measurements.** For KORA, the SOMAscan platform was used to quantify protein levels. It has been described in detail before[5,10,61–65]. Briefly, undepleted EDTA-plasma is diluted into three dilution bins (0.05, 1, 40%) and incubated with bin-specific collections of bead-coupled SOMAmers in a 96-well plate format. Subsequent to washing steps, bead-bound proteins are biotinylated and complexes comprising biotinylated target proteins and fluorescence-labelled SOMAmers are photocleaved off the bead support and pooled. Following recapture on streptavidin beads and further washing steps, SOMAmers are eluted and quantified as a proxy to protein concentration by hybridization to custom arrays of SOMAmer-complementary oligonucleotides. Based on standard samples included on each plate, the resulting raw intensities are processed using a data analysis work flow including hybridization normalization, median signal normalization and signal calibration to control for inter-plate differences. One-thousand blood samples from the KORA F4 study were sent to SomaLogic Inc. (Boulder Colorado, USA) for analysis. Two of the shipped samples were incorrectly pulled from the bio-bank and had no corresponding genotype data, one sample failed SOMAscan QC, leaving a total of 997 samples from 483 males and 514 females for analysis. Data for 1,129 SOMAmer probes (SOMAscan assay V3.2) was obtained for these samples. Five of the probes failed SOMAscan QC, leaving a total of 1,124 probes for analysis (Supplementary Table 1).

Three-hundred and fifty-two samples from QMDiab were analysed at the WCM-Q proteomics core, 338 of which overlapped with samples that were also genotyped. No samples were excluded. Protocols and instrumentation were provided and certified using reference samples by SomaLogic Inc. Experiments were conducted under supervision of SomaLogic personnel. No samples or probe data was excluded.

**GWAS discovery study.** We used PLINK (version 1.90b3w, Shaun Purcell, Christopher Chang, https://www.cog-genomics.org/plink2) (ref. 66) to fit linear models to inverse-normalized probe levels, using age, gender and body mass index as covariates. R version 3.1.3 (R: A language and environment for statistical computing, R Foundation for Statistical Computing, Vienna, Austria, 2015, http://www.R-project.org/) was used for data organization, plotting and additional statistical analyses outside of the actual GWAS, including computation of linear regression models based on raw and log-scaled data. We retained all SNP-probe associations with uncorrected $P$ values of a linear regression with genotype $<10^{-5}$ (14,647 associations covering 3,169 unique SNPs and 1,118 unique probes, Supplementary Data 1, see Supplementary Fig. 7 for a Manhattan plot with all associations). Genomic inflation was low (mean = 1.0035, max = 1.0251). Therefore no correction for genomic control was applied. To define genetically independent association loci, we lumped in a first step for every given probe all associations with correlated SNPs (linkage disequilibrium $r^2 > 0.1$, window size

10 Mb) and retrieved the SNP that had the strongest association at the sentinel for this probe association. We then grouped all highly correlated sentinel SNPs (linkage disequilibrium $r^2 > 0.9$, window size 10 Mb) into single loci. We report uncorrected association $P$ values from linear regression throughout this paper. In all statistical analyses, we require a nominal significance level of 0.05 (alpha error 5%). Multiple hypotheses testing is accounted for by using conservative Bonferroni correction of the significance level, based on the number of tested SNPs (509,946) and probes (1,124), resulting in a genome- and proteome-wide significance level of $P < 8.72 \times 10^{-11}$ (0.05/509,946/1,124). Four-hundred and fifty-one loci had at least one Bonferroni significant sentinel association. For each of these 451 loci, we also considered all probe associations with that same SNP at a significance level of $P < 9.86 \times 10^{-8}$ (0.05/451/1,124) as Bonferroni significant. This resulted in a total of 539 genetic association signals (284 unique probes), each represented by a sentinel SNP and one or more sentinel probes, including 391 *cis*-associations (defined by a SNP closer than 10 Mb from the gene boundaries, 202 unique probes, 18.5% of all 1,090 autosomal probes) and 148 *trans*-associations (96 unique probes, 8.5% of all 1,124 probes). Box-whisker plots and association statistics using alternative data scaling methods (raw, log-normal, inverse-normal) are provided as Supplementary Fig. 8. Annotations of the genetic loci using the SNiPA web-server are provided as Supplementary Fig. 9. Regional association plots are in Supplementary Fig. 6.

**Replication.** Using the QMDiab proteomics data as dependent variables, linear regression models were fitted using PLINK (version 1.90, version b3w), with age, sex, body mass index, diabetes state, the first three principal components of the genotype data and the first three principal components of the proteomics data as covariates. The first three genetic principal components separate the three major ethnicities and the three proteomics principal components account for variability introduced by a low degree of cell haemolysis. Genomic inflation was low (mean = 1.020, max = 1.116). Therefore, no correction for genomic inflation was applied. A tag SNP for replication of each of the 451 Bonferroni significant loci was selected by using the strongest association among all imputed SNPs in the discovery study which had a correlation of $r^2 > 0.8$ with the sentinel SNP. In all, 387 tag SNPs for the 451 originally identified SNPs could be identified for replication, covering 462 SNP-probe pairs out of the originally identified 539 SNP-probe pairs. To consider an association as replicated, we, therefore, required $P < 1.08 \times 10^{-4}$ (0.05/462). 234 (50.6%) of the 462 attempted replications fully replicated at a Bonferroni level of significance ($P < 0.05/462$), 384 (83.1%) displayed nominal significance ($P < 0.05$). Out of 84 non-replicated associations that were nominally significant and that had a MAF <30% in both cohorts, only one association displayed a discordant trend. We estimated the statistical power for the replication by sampling: For each association we randomly selected without replacement 338 individuals from the KORA cohort and computed the $P$ value for a linear regression with genotype on that subset. We repeated this 100 times and report the 5th largest $P$ value from this empirical distribution as the $P$ value that can be expected to be obtained with 95% power (p95). Based on this analysis, we found that 208 SNP-probe pairs with a suitable tag SNP in QMDiab had 95% replication power (p95 $< 1.08 \times 10^{-4}$). 171 of these 208 (82.2%) fully replicated in QMDiab, 198 (95.2%) displayed at least nominal significance.

**Replication of previous SOMAscan based *cis*-associations.** Using the SOMAscan platform on 100 samples, Lourdusamy et al.[5] reported 60 *cis*-associations at a false discovery rate of 5%. Forty-eight of these associations had a suitable tag SNP in the genotyped KORA data set ($r^2 > 0.5$) or association data on an imputed SNP that allowed for replication. Thirty-four of these SNPs were replicated ($P < 0.05/120$, conservatively accounting for replication attempts on tagged and imputed SNPs). The first non-replicated association had rank 21 (Supplementary Table 3).

**Replication of immunoassay based *cis*-associations.** Kim et al.[7] reported 28 *cis*-associations for 27 analytes in the ADNI cohort using plasma proteomic data by multiplex immunoassay on the Myriad Rules Based Medicine (RBM) Human DiscoveryMAP panel v1.0 on the Lumine × 100 platform. Out of 17 associations that had overlapping probes, thirteen were replicated ($P < 0.05/17$) (Supplementary Table 4). Melzer et al.[4] tested 40 protein levels determined by immunoassay and reported 8 pQTLs. We replicated their second strongest association with IL6R. Their strongest association is at the *ABO* locus with TNFa. However, TNFa is not on the SOMAscan panel. We also found numerous other strong associations at the *ABO* locus. Two weak associations of Melzer et al. (SHBG and CRP) were not significant in our study. The remaining associations from Melzer et al. were not targeted by our panel. Enroth et al.[9] used a multiplexed immunoassay and reported 23 *cis*-pQTLs. Two of the proteins (four associations) targeted by their assay were not covered by our panel (VEGF-D, Ep-CAM) and one was located on the X-chromosome and not tested for *cis*-association here (CD40-L). Of the remaining 18 *cis*-pQTLs, twelve associations were replicated ($P < 0.05/18$) (Supplementary Table 5).

**Replication of MS-based *cis*-associations.** Johansson[6] reported five associations and we found *cis*-pQTLs for four of them (Haptoglobin (HP), Alpha-1-antitrypsin

(SERPINA1), Alpha-2-HS-glycoprotein, and APOE isoform 2). We did not, however, find a *cis*-pQTL for Complement C3 (C3), despite the fact that several isoforms were targeted by our panel. Liu *et al.*[8] reported 13 statistically significant association of MS-derived protein levels. Four of their proteins (FCN2, ITIH4, KNG1, PON1) were targeted by our panel, and for two of them we found a *cis*-pQTL in our study (KNG1, FCN2). Wu *et al.*[53] reported protein associations using MS in lymphoblastoid cell lines. However, this study was targeting intracellular proteins and had no overlap with blood circulating proteins targeted by our panel and could, therefore, not be used for replication.

**Proteome annotation.** We used SOMAmer probe identifiers as primary protein identifiers (Supplementary Table 1). Some SOMAmer targets map to multiple Uniprot identifiers ($N = 41$). They either refer to protein complexes ($N = 32$) that are encoded at multiple genome loci, or to different variants of a same protein encoded at a single gene locus ($N = 9$). In all, 1,090 SOMAmer targets were encoded on autosomal chromosomes, 30 targets were encoded on the X-chromosome, and four targeted viral proteins. Genome positions for all SOMAmer targets were retrieved from Ensembl (http://www.ensembl.org) using probe-specific Uniprot identifiers provided by SomaLogic. We used Ingenuity Pathway Analysis (IPA) (http://www.ingenuity.com/products/ipa) to retrieve additional information related to each probe. IPA provides a rich expert-curated knowledgebase of literature-based protein-related information and requires unique mapping to protein identifiers. Forty-one probes were present that target proteins with multiple Uniprot IDs, and these were excluded from the IPA analysis. A further 29 probes were excluded because multiple probes target a same protein. Eight Uniprot IDs could not be mapped by IPA (LAG-1, LD78-beta, NKG2D, HSP 70, and four viral proteins). Ultimately, 1,045 probes with unique Uniprot IDs remained for the IPA annotation.

**Functional annotation of the associations.** We used the SNiPA server (v3.1, http://snipa.org) to annotate 435 out of our 451 lead SNPs (Supplementary Fig. 9). Sixteen SNPs were not available in the SNiPA database (poly-allelic variants are currently excluded). SNiPA provides annotations for all SNPs that are in linkage disequilibrium ($R^2 > 0.8$) with and no more than 500 kb distant from a sentinel SNP using genome assembly data based on GRCh37.p13, Ensembl version 82, 1000-Genomes (phase 3, version 5) data, and GTEx (release 4) eQTL associations for 13 tissues (see columns SNIPA_... in Supplementary Data 1). SNiPA also provides primary effect predictions using the Ensembl VEP tool[67] and all GWAS association data from the GWAS catalogue[68], metabotype associations (http://gwas.eu) and dbGaP (columns GWAS_TRAITS, mGWAS_TRAITS, dbGaP_TRAITS, accessed October 2015). Updates can be retrieved online at http://snipa.org using the block-annotation tool.

**Methylation GWAS.** Data was downloaded from the BIOS QTL browser[69] http://genenetwork.nl/biosqtlbrowser (accessed 9 Feb. 2016). Tagging SNPs ($r^2 > 0.8$) were identified using SNP–SNP correlations from KORA imputed genotypes and meQTLs with BIOS-$P$ values $< 10^{-9}$ were retrieved.

**Metabolomics GWAS.** GWAS associations with mQTLs were obtained using the GWAS-server (http://gwas.eu). This web-server provides access to raw association data from the studies by Suhre *et al.*[1], Shin *et al.*[20] and Raffler *et al.*[70]. Extracted associations were limited to mQTL-$P$ values $< 10^{-8}$ and further requiring $P$-gain $> 10^4$ for ratios[44] (Supplementary Table 8).

**Disease GWAS lookup.** We used SNiPA to identify overlapping disease-GWAS entries. We further downloaded publically available association data for 70 clinically relevant traits from the web-sites of 14 large disease GWAS consortia, based on a list of available GWAS, and downloaded from https://www.med.unc.edu/pgc/downloads. Supplementary Data 4 provides a list with links to all downloaded files and the corresponding publications.

**QMDiab total plasma N-glycosylation.** Unthawed aliquots of identical samples as for the proteome analysis were sent to Genos Ltd. (Zagreb, Croatia) for analysis using ultra-performance liquid chromatography and liquid chromatography mass spectrometry glycoprofiling as follows: Total plasma *N*-glycan release and labelling. Glycans were released from total plasma proteins and labelled as described previously[71]. Briefly, 10 µl plasma sample was denatured with the addition of 20 µl 2% (w/v) SDS (Invitrogen, USA) and *N*-glycans were released with the addition of 1.2 U of PNGase F (Promega, USA). The released *N*-glycans were labelled with 2-aminobenzamide (Sigma-Aldrich, USA). Free label and reducing agent were removed from the samples using hydrophilic interaction liquid chromatography solid-phase extraction. 0.2 µm 96-well hydrophilic polypropylene filter-plate (GH Polypro, Pall Corporation, USA) was used as stationary phase. Samples were loaded into the wells and after a short incubation washed $5 \times$ with cold 90% acetonitrile (ACN). Glycans were eluted with $2 \times 90$ µl of ultrapure water after 15 min shaking at room temperature, and combined eluates were stored at $-20$ °C until use. Total plasma *N*-glycome UPLC analysis. Total plasma *N*-glycans

were analysed by hydrophilic interaction ultra-performance liquid chromatography (HILIC-UPLC) as described previously[71]. Briefly, fluorescently labelled *N*-glycans were separated on an Acquity UPLC instrument (Waters, USA) using excitation and emission wavelengths of 250 and 428 nm, respectively. Labelled *N*-glycans were separated on a Waters BEH Glycan chromatography column, $150 \times 2.1$ mm i.d., 1.7 µm BEH particles, with 100 mM ammonium formate, pH 4.4, as solvent A and acetonitrile (ACN) (Fluka, USA) as solvent B. The separation method used a linear gradient of 30-47% solvent A at flow rate of 0.56 ml min$^{-1}$ in a 23 min analytical run (Supplementary Fig. 10).

**mRNA sequencing and allele specific transcription analysis.** Lappalainen *et al.*[72] report mRNA sequencing of 462 lymphoblastoid cell lines of the 1,000 Genomes Project. RNA sequencing data was downloaded from EBI (http://www.ebi.ac.uk/Tools/geuvadis-das/). Analysis was limited to 373 samples of European ancestry. We used CLCBio genomics workbench (Qiagen Inc.) to align reads, calculate RKPM (Reads Per Kilobase of transcript per Million mapped reads) values and analyse the data.

**Gaussian Graphical Network (GGM) construction.** Using raw (unscaled) data, regressing out age + gender + bmi (997 samples with data for 1,124 variables) we computed a GGM using the ggm.estimate.pcor function from the R GeneNet package. The estimated optimal shrinkage intensity lambda (correlation matrix) was 0.187. We obtained 3,943 GGM edges connecting 1,092 protein nodes with Bonferroni significant partial correlation coefficients ($P < 7.9 \times 10^{-8} = 0.05/(1,124*1,123/2)$), provided as Supplementary Data 3. We added SNP-probe association edges connecting 451 genetic loci to 539 proteins. We further added SNP-disease association edges using all associations reported in the GWAS catalogue (identified using SNiPA at linkage disequilibrium $r^2 > 0.8$). We also added all SNP-disease associations from 84 clinically relevant traits of 14 large GWAS consortia that had a GWAS-$P$ value $P < 10^{-8}$. These SNP-disease edges were further manually curated to ascertain unique SNP-disease pairs. SNP association to disease-related protein levels were excluded.

**Data availability.** All summary statistics and association data are freely available (Supplementary Data 5), accessible online on an integrated web-server at http://proteomics.gwas.eu. A fully functional docker-based version of the web-server can also be freely downloaded from this link for local installation and network-free usage (HTML5-based, no extra software is required). The informed consent given by the study participants does not cover posting of participant level phenotype and genotype data in public databases. However, data for KORA are available upon request from KORA-gen (http://epi.helmholtz-muenchen.de/kora-gen). Requests are submitted online and are subject to approval by the KORA board.

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

## Acknowledgements

This work was supported by 'Biomedical Research Program' funds at Weill Cornell Medicine in Qatar, a program funded by the Qatar Foundation. The statements made herein are solely the responsibility of the authors. M. Arnold was supported by the Helmholtz cross-program topic 'Metabolic Dysfunction'. D. Mook-Kanamori was supported by Dutch Science Organization (ZonMW-VENI Grant 916.14.023). The KORA study was initiated and financed by the Helmholtz Zentrum München—German Research Center for Environmental Health, which is funded by the German Federal Ministry of Education and Research (BMBF) and by the State of Bavaria. Furthermore, KORA research was supported within the Munich Center of Health Sciences (MC-Health), Ludwig-Maximilians-Universität, as part of LMUinnovativ. The KORA-Study Group consists of A. Peters (speaker), J.Heinrich, R.Holle, R.Leidl, C.Meisinger, K.Strauch and their co-workers, who are responsible for the design and conduct of the KORA studies. We gratefully acknowledge the contribution of all members of field staff conducting the KORA F4 study. We thank the staff from HMC dermatology department and WCM-Q clinical research core for their contribution to QMDiab. We thank Brian Sellers from SomaLogic for support with measuring the QMDiab samples at WCM-Q. We acknowledge free access to summary statistics provided by the GWAS consortia listed in Supplementary Data 4. We thank Jenine Davidson for the design of Figure 1a. Most of all, we thank all study participants of KORA and QMDiab for their invaluable contributions to this study.

## Author contributions

Jointly supervised research: K.S., J.G., C.G. Conceived and designed the experiments: K.S., J.G., C.G. Performed the experiments: R.E., J.G., E.K.A., Y.A.M., J.M., H.S., G.L., M.P., Performed statistical analysis: K.S., C.G., A.W. Analysed the data: K.S., A.M.B., R.J.C., J.G., G.T., M.A., C.G., G.K., J.R., A.W. Contributed reagents, materials, or analysis tools: K.S., M.A.S., M.A., H.G., G.K., A.P., J.R., K.S., D.O.M., R.K.D., L.G. Wrote the paper: K.S. All authors discussed the results and reviewed the final manuscript.

## Additional information

**Competing financial interests:** M.P., G.L., K.D. and L.G. are working for or have stakes in Genos Ltd. and Somalogic Inc., respectively. The remaining authors declare competing no financial interests.

