## [Peer Review File · Nature Communications]

Reviewers' Comments:

Reviewer #1 (Remarks to the Author)

In this manuscript, Suhre and colleagues provide the most detailed description yet of the genetics of individual variation in plasma proteome levels, involving Somalogic aptamer assays of ~1100 proteins in ~1300 individuals. These are important data given growing expectation about the value of this particular technique in enabling relatively high throughput analysis of the proteome. Several other groups have generated data on a similar scale and must be preparing equivalent publications. As the authors show, the pick-up rate in terms of significantly associated pQTLs is substantial. They provide multiple examples of the ways in which these pQTL discoveries help to provide insights into disease mechanisms, to deconvolute GWAS signals, and to highlight novel therapeutic opportunities.

The manuscript is extensive with abundant supplementary data. The associations identified and the protein networks they have characterised are made available through a website, and the authors have undertaken to make this open access at publication for which they are to be commended. (However, I found the website was not terribly robust and kept crashing on me when I accessed it).

The manuscript is relatively clear. The more "genomic" aspects are dealt with rather briefly, and most of the results are given over to exemplifying the value of these data in setting up mechanistic hypotheses that can direct further empirical study. This reviewer would have liked to see a more extensive description of the genomic aspects (e.g. more details on the cis vs trans comparisons, on the numbers of cis effects seen (with appropriately liberal p values), on the overlap with eQTL data from various tissues) but perhaps that is destined for another manuscript. The multiple examples provided are interesting, and do serve to illuminate the value of these data, but they are at the same time, essentially speculative and descriptive.

I have some concerns. Amongst the main ones are

- * failure to confirm directional consistency between primary and replication samples. I assume that the majority of the replication is directionally consistent given the high proportion of primary signals that replicate, but this should be stated more clearly
- * a reliance on "overlap" of signals (eg between pQTLs and eQTLs, between pQTLs and disease GWAS signals) rather than confirming that those overlaps are likely to reflect the same locus rather than two distinct loci that map to the same approximate region. (As more and more eQTLs and pQTLs and GWAS loci are found, this is more and more of an issue, and failure to take this into account can lead to false attribution of biological connections, where none is justified)

Other issues:

- * what measures were used to account for hidden structure (eg using PEER or equivalent) or was this not deemed necessary?
- * the trans threshold is certainly conservative, overly so for attribution of cis effects. Please provide some additional insights into the detection of cis effects (recognising of course that only 5% of protein coding genes are assayed).
- * line 89: the authors point out elsewhere that cis effects may be artefacts related to allele-specific aptamer binding effects. Is the count of coding variants cis-pQTLs (228/384) greater than expected by chance
- * line 87 and elsewhere: which ethnicity is used to derive r^2 measures (given that the Qatari replication set is not European)
- * line 127-155: the hypotheses here are not clearly stated. Some seems overly speculative (e.g. line 133 has multiple qualifiers; "suggest" "potential")
- * line 212: a propos of comments above regarding overlaps between QTLs, the statement here is

hard to follow. If rs10801582 tags rs4915318 which is "an eQTL for CHFR4 expression in liver", then surely rs10801582 is also presumably an eQTL for CHFR4 too!

Reviewer #2 (Remarks to the Author)

The authors present a large-scale analysis of protein levels in the blood of two distinct populations. They obtained genotype data for the same individuals and performed GWA of the levels of 1124 proteins. They report a series of interesting associations between genotypes, diseases and protein levels.

The manuscript jumps about from one topic to the next without a clear flow of ideas. I understand that there is a lot here, but it might be worth focusing on a few key results, perhaps that relate to autoimmune or complement factor findings.

The authors mention that the effect of genetic variation may affect aptamer binding and hence reported protein levels (line 318-327). This seems like a critical issue. On lines 323-4, they mention "the absence of correlated SNPs that alter the protein structure (e.g., based on SNIPA-annotated 1000 Genomes data)". I may have missed it in the Methods, but did the authors scan each protein for non-synonymous SNPs in the genetic sequence of each measured protein? It would be good to add this sentence along with the result in the Methods.

Along these same lines, how might alternative splicing between individuals affect protein measurements?

Minor: Why is there color in Figure 4B?

The data is available as described in the manuscript and the associated web link is live.

Reviewer #3 (Remarks to the Author)

Surhe et al have "married" protein data (based on an aptamer platform) with genetic information in derivation and validation cohorts – a strength of this manuscript. The initial analyses were performed in the KORA cohort, with validation in QMDiab. There is a very comprehensive annotation of pQTLs using SNIPA, pathway analyses using IPA, Gaussian network modeling, and publication of summary data on the web.

However, there were two very large problems with the manuscript as it is currently presented. First, the ability of trans-pQTLs to identify "causal links" between genetic variants and circulating proteins is significantly—and repeatedly—overstated. It is certainly very possible that the many trans-associations highlighted in the paper are related not through a shared biological pathway as suggested in the manuscript, but rather through a shared common phenotype. This possibility is perhaps best exemplified using the authors' discussion of the ABO locus in the body of the paper and in Supplemental Table 13, which describes the many clinical phenotypes that have previously been associated with the pleiotropic ABO locus. The finding that circulating levels of the insulin receptor, for example, are correlated with the ABO locus could very well reflect the well-established independent associations between the ABO locus and diabetes and the insulin receptor and diabetes. It is not possible to place the insulin receptor in a "causal" pathway connecting the ABO-diabetes association as the authors suggest, given the currently presented data. This is true for several (if not all) of the "hypotheses" presented in the Trans-association section of the results section. This is also true of many of the hypotheses presented throughout the remainder of the results section. For example, it is not convincing that the current data is able to place GP33 "in the pathway between" the FUT3 locus and circulating levels of CA19-9, as the authors suggest. These shared associations rather place GP33, the FUT3 locus, and CA19-9 protein levels in a similar network, which may reflect shared clinical phenotypes or other shared network members. The

authors partially address this concern by pointing out that several of these trans-associations (< 1/2) were successfully replicated in the QMDiab cohort with different baseline clinical characteristics from KORA, however, one cannot equate these associations to causal pathways without further direct experimentation. Ideally, at least one of the biological findings would have been followed up with functional data of some kind. The manuscript would be significantly stronger if the authors presented the GWAS analyses, pQTL annotations, pathway analyses, and network modeling in the nicely presented online tool that they have developed and moved discussions of possible biological pathway relevance (pending further direct experimentation) entirely to the discussion and supplemental figure(s). As it stands, the manuscript reads more like bullet points written by different writers. Unfortunately, the integration of speculation and discussion in the results sections makes the manuscript almost impenetrable to read.

A second problem is the need for a more complete discussion accounting for the many proteins that were not found to have associations with cis regions of their encoding genes. This raises potential concerns with regards to aptamer specificity for the intended target protein. The authors include some analyses detailing replication of cis-pQTLs from GWAS studies with reported ELISA assays; it would be perhaps more valuable to include similar analyses specifically for proteins that associated only with trans-pQTLs (as these may be proteins that were mis-identified by aptamer-affinity assays). Were any of the findings followed up by mass spec??? The specificity of this platform has been questions in a variety of recent manuscripts. Lastly, the authors raise the potential concern that identified cis-acting genetic variants could, in theory, interrupt target protein-aptamer affinity. It would be very interesting to potentially include some more direct analyses of this possibility (?again MS/MS or other data?), though the authors did present several lines of indirect evidence including allele-specific analyses and eQTL data addressing this concern.

"Connecting genetic risk to disease endpoints through the human blood plasma proteome"

Reviewer #1:

In this manuscript, Suhre and colleagues provide the most detailed description yet of the genetics of individual variation in plasma proteome levels, involving Somalogic aptamer assays of ~1100 proteins in ~1300 individuals. These are important data given growing expectation about the value of this particular technique in enabling relatively high throughput analysis of the proteome. Several other groups have generated data on a similar scale and must be preparing equivalent publications. As the authors show, the pick-up rate in terms of significantly associated pQTLs is substantial. They provide multiple examples of the ways in which these pQTL discoveries help to provide insights into disease mechanisms, to deconvolute GWAS signals, and to highlight novel therapeutic opportunities.

Response: We thank the reviewer for his/her constructive comments.

The manuscript is extensive with abundant supplementary data. The associations identified and the protein networks they have characterised are made available through a website, and the authors have undertaken to make this open access at publication for which they are to be commended. (However, I found the website was not terribly robust and kept crashing on me when I accessed it).

Response: We are sorry to hear that the reviewer had issues with the server. We are constantly monitoring the proper functioning of the website, and will immediately fix any issue that comes to our attention. All crashes that we could identify so far were related to network connectivity problems outside of our network. Please note that we also provide the entire website as a docker image for free download. It is based on modern HTML5 code and avoids any dependencies on exotic third-party plugins. It can therefore easily be installed on a local computer and run under Linux, Windows or Apple iOS operating systems in a closed network, which may be of particular interest for users working in private industry.

The manuscript is relatively clear. The more "genomic" aspects are dealt with rather briefly, and most of the results are given over to exemplifying the value of these data in setting up mechanistic hypotheses that can direct further empirical study. This reviewer would have liked to see a more extensive description of the genomic aspects (e.g. more details on the cis vs trans comparisons, on the numbers of cis effects seen (with appropriately liberal p values), on the overlap with eQTL data from various tissues) but perhaps that is destined for another manuscript. The multiple examples provided are interesting, and do serve to illuminate the value of these data, but they are at the same time, essentially speculative and descriptive.

Response: We agree with the reviewer that a more extensive discussion of the genomic aspects would be interesting. Given the limited space and the high number of biologically relevant associations, we decided to focus this paper more on the biological aspects of the associations and less on the genomic ones. We provide information on the overlap with eQTL data from various tissues, data that allows cis vs trans comparisons, and on cis effects with more liberal p values, both on the web server and as Supplemental Tables. However, the SOMAscan assay is a targeted platform that covers a selection of proteins with biases towards proteins of biological interest, feasibility of developing sensitive and specific

aptamers, and detectability of the proteins in blood. Genomic statistics would be affected by this selection bias. We therefore chose not to discuss these numbers in the paper.

I have some concerns. Amongst the main ones are

* failure to confirm directional consistency between primary and replication samples. I assume that the majority of the replication is directionally consistent given the high proportion of primary signals that replicate, but this should be stated more clearly

Response: The reviewer is correct in assuming that the majority of the replication is directionally consistent: Out of 234 replicated SNPs, 215 displayed directional consistency between primary and replication sample. The 19 SNPs that displayed opposing directionality had a minor allele frequency on average of 45.3% (minimum 37.5%), most of which can be explained by a flip in major and minor allele coding between the two cohorts. Only 3 of the 19 directionally inconsistent SNPs were on a same SNP and had identical alleles (locus 282: Spondin-1, locus 350: TECK, locus 367: MICB). Directional inconsistency for these three SNPs may potentially be a result of differences in the genetic LD structure, which may also lead to a flip in major and minor allele coding for a tag SNP. We added the following sentence to the manuscript: "We observed directional consistency between primary and replication sample for 215 of the 234 replicated SNPs. Discrepancy for the remaining 19 SNPs may be explained by changes in major and minor allele coding between the two cohorts and ethnic differences."

* a reliance on "overlap" of signals (eg between pQTLs and eQTLs, between pQTLs and disease GWAS signals) rather than confirming that those overlaps are likely to reflect the same locus rather than two distinct loci that map to the same approximate region. (As more and more eQTLs and pQTLs and GWAS loci are found, this is more and more of an issue, and failure to take this into account can lead to false attribution of biological connections, where none is justified)

Response: In our definition of overlapping signals we do not rely on physical proximity of the associated variants, but require that the SNPs that underlie two quantitative trait associations are in strong linkage disequilibrium ($r^2 > 0.8$). We specify this in the methods section by writing: "SNiPA provides annotations for all SNPs that are in linkage disequilibrium ($R^2 > 0.8$) with and no more than 500kb distant from a sentinel SNP using genome assembly data based on GRCh37.p13, Ensembl version 82, 1000-Genomes (phase 3, version 5) data, and GTEx (release 4) eQTL associations for 13 tissues (see columns SNIPA_... in Supplemental Table 3)." The limitation to search for overlapping SNPs within a region of no more than 500kb distant from a sentinel SNP is an additional constraint, imposed to limit the computation of pairwise SNP correlations to a reasonable dimension.

Other issues:

* what measures were used to account for hidden structure (eg using PEER or equivalent) or was this not deemed necessary?

Response: For the KORA study, we do not deem it necessary to account for hidden population structure. The KORA population is genetically very homogeneous with participants living in the region of Augsburg in Southern Germany. The mean genomic inflation in KORA is 1.0035, and the maximal observed inflation is 1.0251. QMDiab is a mix of different ethnicities. To correct for population heterogeneity, we included the first three principal components of the genetic data (computed using PLINK) as covariates in the

model. After inclusion of the three PCs, mean genomic inflation in QMDiab was 1.020, with a maximum of 1.116. We describe this in the methods section as follows:

KORA: "Genomic inflation was low (mean=1.0035, max=1.0251). Therefore no correction for genomic control was applied."

QMDiab: "Using the QMDiab proteomics data as dependent variables, linear regression models were fitted with PLINK (version 1.90, version b3w), using age, sex, BMI, diabetes state, the first three principal components (PCs) of the genotype data and the first three PCs of the proteomics data as covariates. The first three genetic PCs separate the three major ethnicities and the three proteomics PCs account for variability introduced by a low degree of cell haemolysis. Genomic inflation was low (mean=1.020, max=1.116). Therefore no correction for genomic inflation was applied."

* the trans threshold is certainly conservative, overly so for attribution of cis effects. Please provide some additional insights into the detection of cis effects (recognising of course that only 5% of protein coding genes are assayed).

Response: Limiting the set to the 1,045 aptamers that have unique Uniprot IDs and that can be mapped using IPA, and defining a cis association by a SNP that is closer than 1MB to either gene start or end, we find 3,064 cis associations for 220 unique proteins at a p-value cut-off $< 5 \times 10^{-8}$. (Bonferroni correction assuming that there are 1,000 SNPs every 1MB window around a cis-encoded protein). We added the following sentence to our manuscript: "For 21% of the assayed proteins we detected a cis-pQTL at a more liberal significance cut-off of $p < 5 \times 10^{-8}$."

* line 89: the authors point out elsewhere that cis effects may be artefacts related to allele-specific aptamer binding effects. Is the count of coding variants cis-pQTLs (228/384) greater than expected by chance

Response: A genetic variant that changes the protein surface may in principle change the binding efficiency of an aptamer, without affecting the protein's concentration. Hence, such "epitope effects" may lead to a misinterpretation of a cis-association signal as a change in protein concentration, where in fact only the aptamer binding is affected. This is an issue that is rarely mentioned, but equally present in all immune-assay experiments, and in a different way also in MRM based mass spec technologies (by changing fragment mass).

There are several ways to rule out such epitope effects (or at least reduce their likelihood), as we discuss in our paper: allele-specific gene expression, overlapping eQTLs, multiple independent association signals, and replication using another technique. We give examples for all of these.

An additional argument to rule out an epitope effect is the absence of any epitope changing SNP linked to the pQTL. We describe the result of this analysis as follows: "Of 384 annotated cis-pQTLs, 228 had a variant in the gene coding region, whereof 74 were protein-changing." With this statement we intend to say that only 74 out of 384 cases could potentially be affected by a direct epitope effect, although some of the non-coding variants among the 228 SNPs in the coding region may potentially affect splice sites and lead to an epitope effect. To make this clear we write in the discussion: "Genetic variance in a protein sequence may affect its higher order structure, and thus, its aptamer-binding affinity. [...] These

issues remain to be addressed. [...]Most importantly, trans-associations, which were a central focus of this study, are not affected by this type of potential artefact.”

Regarding the question of whether the count of coding variants cis-pQTLs (228/384) is greater than what would be expected by chance, we found no easy-to-define null-model that would allow to test this hypothesis. The 228 coding variants are in LD with GWAS selected SNPs, and the 384 proteins were selected for by the SOMAScan assay. It is hence hard to define a randomization experiment that is bias-free. Also, we are not sure what outcome we would actually expect (probably it should be non-significant?) and how to interpret a result that turns out to be significant (how would we know that this is not due to the wrong null-model?). We therefore did not attempt to compute a p-value for this question.

* line 87 and elsewhere: which ethnicity is used to derive r2 measures (given that the Qatari replication set is not European)

Response: r2 measures were derived using different ethnicities, depending on the situation: To relate SNPs within individual studies to each other, we used r2 values computed using the actual genotype data from KORA and QMDiab, respectively. To identify proxy SNPs for replication between the two studies, r2 values were computed using the imputed KORA data. Please note that we did not impute the QMDiab data, since we expect this to be a more complex issue due to the mixed ethnicities and limited availability of haplotypes from the Middle East and the Philippines. Failure to identify the best replication SNP by using r2 from a non-related population can potentially lead to non-replicated pQTLs. However, in our study we observed only very few strong but non-replicated pQTLs. Manual inspection of these cases (using data from a GWAS on the QMDiab protein data) showed that we did not miss any replication due to differences in genetic structure between ethnicities.

* line 127-155: the hypotheses here are not clearly stated. Some seems overly speculative (e.g. line 133 has multiple qualifiers; "suggest" "potential")

Response: We agree with the reviewer, hypotheses should not be expressed with additional “hypothetical” qualifiers. Our intention was not to sound overly speculative here. The data actually supports these hypotheses. We modified the text accordingly (see marked-up word document)

* line 212: a propos of comments above regarding overlaps between QTLs, the statement here is hard to follow. If rs10801582 tags rs4915318 which is "an eQTL for CHFR4 expression in liver", then surely rs10801582 is also presumably an eQTL for CHFR4 too!

Response: Yes, this phrase was not clearly stated: rs10801582 is of course also an eQTL for CHFR4. This is what we intended to convey. We now write: “It [the rs10801582 variant] is an eQTL for CFHR4 expression in liver, as it tags the GTEx rs4915318 variant (GTEx, $p=1.6 \times 10^{-7}$).

Reviewer #2:

The authors present a large-scale analysis of protein levels in the blood of two distinct populations. They obtained genotype data for the same individuals and performed GWA of the levels of 1124 proteins. They report a series of interesting associations between genotypes, diseases and protein levels.

Response: We thank the reviewer for his/her constructive comments.

The manuscript jumps about from one topic to the next without a clear flow of ideas. I understand that there is a lot here, but it might be worth focusing on a few key results, perhaps that relate to autoimmune or complement factor findings.

Response: We agree with the reviewer that a paper describing results from a GWAS sometimes appears to jump about from one topic to the next. This resides in the nature of the non-targeted GWAS approach and makes reading (and writing) such papers a bit more challenging than that of a neat story of a hypothesis-driven investigation. We feel that by limiting our paper to isolated results we would curtail the broader interest of our paper. We acknowledge that we provide a mosaic of ideas here, but we strived to present them in such a way that the reader obtains one example for every general topic, highlighting how to best use our data for their own research.

The authors mention that the effect of genetic variation may affect aptamer binding and hence reported protein levels (line 318-327). This seems like a critical issue. On lines 323-4, they mention "the absence of correlated SNPs that alter the protein structure (e.g., based on SNIIPA-annotated 1000 Genomes data)". I may have missed it in the Methods, but did the authors scan each protein for non-synonymous SNPs in the genetic sequence of each measured protein? It would be good to add this sentence along with the result in the Methods.

Response: We indeed scanned each protein for non-synonymous SNPs in the genetic sequence of each measured protein using the SNIIPA server. SNIIPA identified all potential coding variants that are in high LD with our lead pQTL SNPs. We describe the result of this analysis as follows: "Of 384 annotated cis-pQTLs, 228 had a variant in the gene coding region, whereof 74 were protein-changing."

Along these same lines, how might alternative splicing between individuals affect protein measurements?

Response: Alternative splicing may also alter protein structure and hence aptamer binding. SNPs affecting splice variants were also identified by SNIIPA. For individual proteins these results can be accessed using the online web-tool.

Minor: Why is there color in Figure 4B?

Response: The reviewer is correct in pointing this out. The data presented here comes from an entirely different experiment (mRNA sequencing of cell lines) and represents an independent follow-up to one of our associations – we intended to emphasize this fact by presenting the data differently. However, color is clearly not required to make this point. We therefore changed the figure and now present the data using uniformly gray-shaded boxes.

The data is available as described in the manuscript and the associated web link is live.

Reviewer #3:

Surhe et al have “married” protein data (based on an aptamer platform) with genetic information in derivation and validation cohorts – a strength of this manuscript. The initial analyses were performed in the KORA cohort, with validation in QMDiab. There is a very comprehensive annotation of pQTLs using SNIIPA, pathway analyses using IPA, Gaussian network modeling, and publication of summary data on the web.

Response: We thank the reviewer for his/her constructive comments.

However, there were two very large problems with the manuscript as it is currently presented. First, the ability of trans-pQTLs to identify “causal links” between genetic variants and circulating proteins is significantly—and repeatedly—overstated. It is certainly very possible that the many trans-associations highlighted in the paper are related not through a shared biological pathway as suggested in the manuscript, but rather through a shared common phenotype. This possibility is perhaps best exemplified using the authors’ discussion of the ABO locus in the body of the paper and in Supplemental Table 13, which describes the many clinical phenotypes that have previously been associated with the pleiotropic ABO locus. The finding that circulating levels of the insulin receptor, for example, are correlated with the ABO locus could very well reflect the well-established independent associations between the ABO locus and diabetes and the insulin receptor and diabetes. It is not possible to place the insulin receptor in a “causal” pathway connecting the ABO-diabetes association as the authors suggest, given the currently presented data. This is true for several (if not all) of the “hypotheses” presented in the Trans-association section of the results section. This is also true of many of the hypotheses presented throughout the remainder of the results section. For example, it is not convincing that the current data is able to place GP33 “in the pathway between” the FUT3 locus and circulating levels of CA19-9, as the authors suggest. These shared associations rather place GP33, the FUT3 locus, and CA19-9 protein levels in a similar network, which may reflect shared clinical phenotypes or other shared network members. The authors partially address this concern by pointing out that several of these trans-associations (< 1/2) were successfully replicated in the QMDiab cohort with different baseline clinical characteristics from KORA, however, one cannot equate these associations to causal pathways without further direct experimentation.

Response:

We agree that our use of the term “causality” in the context of trans-pQTLs requires some qualification.

What we intended to say is that, except for very exotic cases, a genetic variant that associates with a disease end point is always causal for the process that leads to an increased disease risk, since the presence of a disease does not change a genetic variant. Moreover what we also intended to say is the following: a genetic variant acts initially in cis, by changing a protein’s properties (coding or splice variants), its transcription level (regulatory variants), or some other cis-encoded, non-protein coding element, such as a miRNA. Genetic variance in such a cis-encoded “effector” would then directly or indirectly lead to a change in the trans-encoded protein, which we observe as a pQTL. Hence, there is causality between the change of a cis-encoded effector and the trans-encoded protein levels we measure. We agree with the reviewer that this association alone does not place the trans-pQTL between the genetic variant and the disease association, since confounding is always a possibility. This is not what we intended to imply.

Our main message, that pQTLs can connect genetic risk loci to disease, has to be understood in the sense of GWAS hypothesis generation. To make this clear, we added the follow statement: “While many of our associations connect these proteins to disease pathway through shared association, it should be borne in mind that confounding is always a possibility that needs to be ruled out by further experimentation.”

Regarding the ABO-INSR association we now write: “The association between ABO and the Insulin receptor (INSR) reflects the well-established associations between the ABO locus and diabetes and the insulin receptor and diabetes. The association between ABO and the Insulin receptor (INSR) suggest that INSR-mediated insulin signaling may be involved in the ABO-diabetes association”.

Regarding the FUT3 association we wrote: “Our data suggested a role for GP33 in the pathway between the variance in the FUT3 locus and the expression of cancer antigen, CA19-9.” Here again, we intended this statement to be understood as a hypotheses generated by a GWAS, that requires further experimentation. We reformulated this statement to “The previously reported association between variance in the FUT3 locus and the expression of cancer antigen, CA19-9, and our observation of a similar association with GP33, suggest that both glycans may be involved in a same pathway.”

In summary, we agree with the reviewer’s criticism that we used the term causality too broadly, and modified the manuscript in several places in order not to imply causality where none is shown (see marked-up version of the paper).

Ideally, at least one of the biological findings would have been followed up with functional data of some kind.

Response: We agree that following up on biological findings from GWAS with functional data would be ideal. For this purpose we actually included the analysis of the glycosylation data (which was specifically generated for this study and has never been published before) as a follow up with functional data (Figure 3). With that data we show that genetic variance in the FUT3 protein levels also leads to variance in its downstream products, namely the GP33 glycans. Furthermore, we feel that the analysis of the mRNA sequencing data in the ERAP1 and CPNE1 cases (albeit taken from public sources) also provides functional follow-up (Figure 4), as they show that the observed multi-SNP pQTL of ERAP1 is matched by a multi-SNP eQTL, and that the association in the CPNE1 case is supported by an allele-specific expression pattern. Hence, we believe our paper to present some follow up with functional data of some of the biological findings.

The manuscript would be significantly stronger if the authors presented the GWAS analyses, pQTL annotations, pathway analyses, and network modeling in the nicely presented online tool that they have developed and moved discussions of possible biological pathway relevance (pending further direct experimentation) entirely to the discussion and supplemental figure(s). As it stands, the manuscript reads more like bullet points written by different writers. Unfortunately, the integration of speculation and discussion in the results sections makes the manuscript almost impenetrable to read.

Response: We agree with the reviewer that hypotheses generated by a GWAS call for further experimentation. The fact that the results we present may read on first view like a bullet point list of hypotheses resides in the nature of the hypotheses-free GWAS approach (new hypotheses are the end product of a GWAS). To make our results more accessible to the reader, we developed the online tool, in order to allow the reader to follow up on individual cases of interest, providing pQTL annotations,

pathway analyses, and network modeling. In the manuscript we present examples of the many different hypotheses for biomedical applications that result from our GWAS findings.

A second problem is the need for a more complete discussion accounting for the many proteins that were not found to have associations with cis regions of their encoding genes.

Response: We politely disagree on this point. We don't think that the many proteins that were not found to have associations within cis regions of their encoding genes represent a problem. This lack of association can be attributed mainly to the lack of statistical power to detect associations with weaker effect sizes in an $N=1000$ GWAS, in addition to the possibility that blood concentrations of some proteins may just not be under genetic control. The plot below shows the p-values of association as a function of the genetic effects beta (data from Supplemental Data 2, zoom on associations with $p > 10^{-30}$). Given the size of our study population and the overall variance in the data, any association with an absolute effect size smaller than 0.25 (s.d./copy number of the minor allele) would not have reached genome-wide significance, and is thus not detected.

Volcano plot: association p-values plotted against effect size (beta = slope of the linear model).

This situation is not different to the case of GWAS with other omics data sets. In the case of gene expression, a GWAS with $N=1000$ would not identify eQTLs for most of the transcripts neither. The same is true for GWAS with metabolomics. In our own $N=300$ mGWAS we identified 4 hits (Gieger et al, PLoS Genetics, 2008). In follow up an with $N=1,800$ we identified 37 mQTL associations (Suhre et al., Nature, 2012), and that number increased to 145 associations in our large $N=6,800$ study (Shin et al., Nature Genetics, 2014). This observation is furthermore in agreement with the fact that Lourdosamy et al. (Human Mol. Genet., 2012) detected only 60 cis-pQTLs in their small $N=100$ pGWAS with the SOMAScan platform (at an FDR of 5%).

To make this clear we added the following sentence to the discussion: “In this study, about 20% of all assayed proteins had a significant pQTL association. This number is expected to increase in future more highly powered studies”

This raises potential concerns with regards to aptamer specificity for the intended target protein. The authors include some analyses detailing replication of cis-pQTLs from GWAS studies with reported ELISA assays; it would be perhaps more valuable to include similar analyses specifically for proteins that associated only with trans-pQTLs (as these may be proteins that were mis-identified by aptamer-affinity assays).

Response: Actually, our analysis of the GWAS studies with reported ELISA includes similar analyses specifically for proteins that associated only with trans-pQTLs: VWF, SELE, SELP, and ICAM1. The header of Supplemental Table 2 (previously Supplemental Table 4) was mislabeled as “Replication of cis-pQTLs from GWAS studies with ELISA assays”. We corrected this error.

Furthermore, as an additional argument to rebut the concern that trans-pQTLs may be proteins that were mis-identified by aptamer-affinity assays, note that there are a number of proteins that have both, a pQTL in trans and in cis. These are MBL, the TLR4/MD-2 complex, CD109, P-selectin, Prekallikrein, Catalase, Coagulation Factor XI, sTie-2, VEGF sR2, EDAR, and DC-SIGN. In the case of P-selectin, both cis and trans pQTLs also replicate previously published immuno-assay based associations. There are also a few loci that harbor cis- and trans-pQTLs together. In some cases the associated proteins are co-regulated (e.g. the Complement C1r and C1q subcomponents) or the cis- and the trans-associated proteins are otherwise functionally related (e.g. haptoglobin and ferritin). We therefore feel that there is no reason to fear that, in general, proteins in trans-pQTLs were mis-identified by the aptamer-affinity assays.

Were any of the findings followed up by mass spec???

Response: We did not directly follow up on our results using mass spectroscopy. Doing so for all associations would be infeasible, and the interest of following up on selected ones is only limited, since the feasibility of such a follow up has already been shown recently by Ngo et al. (Circulation, 134:270–285, 2016; see Figure below). The authors found that response curves for the selected aptamer-enriched proteins were linear over a wide dynamic range of spiked protein concentrations.

Figure 5 from Ngo et al.: Aptamers bind their cognate proteins spiked into plasma. Proteins were spiked into plasma in the presence of biotinylated aptamers bound to streptavidin bead. Following elution and digestion of the affinity-enriched sample, LC-MRM-MS analysis was performed. Shown are MS signal intensities for peptides unique to 8 proteins in the study that were extracted from the total ion chromatogram. Source: <http://circ.ahajournals.org/content/134/4/270.long>

Furthermore, Finkel et al. (results presented as a poster at the Hupo conference) screened eight cancer cell lines by the SOMAscan assay and compared the results to data dependent mass spectrometry profiling using TMT10 on the Q Exactive Plus. Twenty proteins were selected and analyzed in cell lysate by multiple reaction monitoring (MRM) after pull-down using the respective SOMAmer reagent. They found that the SOMAscan and MRM data show consistent correlation across multiple proteins.

Figure from Finkel et al.: SOMAscan and Targeted Proteomics (MRM) results were compared to RNA seq data for proteins detected in at least 3 cell lines.

Source: http://www.somalogic.com/somalogic/media/Assets/Posters/Finkel_Hupo_poster_final.pdf

To make this clear we added the following sentence to the manuscript: “We did not directly follow up on our results using mass spectroscopy. However, Ngo et al. (Circulation, 134:270–285, 2016) recently showed that response curves for selected aptamer-enriched proteins were linear over a wide dynamic range of spiked protein concentrations”

The specificity of this platform has been questions in a variety of recent manuscripts.

Response: While isolated cases of aptamers that display non-specificity may have been published, we are not aware of any publication that questions the aptamer specificity of the SOMAscan platform in general. In addition to the stringent selection process and QC performed by the company itself, we take the identification of the many cis-pQTL is actually a very strong indicator of aptamer specificity. As with every large-scale technology, it is always possible that individual aptamers may have specificity issues, but especially in the light of the numerous pQTLs we replicate from other technologies, we do not think that this is a major issue.

However, to emphasize the reviewer’s concern more, we added the term “SPECIFICITY” to the discussion of our study limitations, which now reads: “Genetic variance in a protein sequence may affect its higher order structure, and thus, its aptamer-binding affinity AND SPECIFICITY. Similarly, alterations in protein structure may affect binding in immunoassay-based methods and protein mass in targeted MS-based methods. These issues remain to be addressed.”

Lastly, the authors raise the potential concern that identified cis-acting genetic variants could, in theory, interrupt target protein-aptamer affinity. It would be very interesting to potentially include some more direct analyses of this possibility (?again MS/MS or other data?), though the authors did present several lines of indirect evidence including allele-specific analyses and eQTL data addressing this concern.

Response: We agree with the reviewer that the concern of cis-acting genetic variants, which could potentially interrupt target protein-aptamer affinity, needs to be addressed. We present several ways to deal with this issue, such as including mRNA sequencing data for allele-specific gene expression. Although not being based on mass spectrometry, we feel that this case provides a direct analysis of this possibility using other data from samples that are independent of our initial study (see Supplemental Figure 5).

While we agree that a follow up on all associations using mass spec may be optimal, we feel that the interest of such an experiment would be of limited interest, since it would only be reasonably feasible for a small subset of proteins. As we state above, experiments comparing the SOMAscan platform to mass spectrometry have already been performed by others on selected proteins.

As mentioned above, we make the reader aware of this caveat by writing: “Genetic variance in a protein sequence may affect its higher order structure, and thus, its aptamer-binding affinity AND SPECIFICITY. Similarly, alterations in protein structure may affect binding in immunoassay-based methods and protein mass in targeted MS-based methods. These issues remain to be addressed.”

Reviewers' Comments:

Reviewer #1 (Remarks to the Author)

This is a revision of a manuscript that I saw in its original version at this journal. I can see that all three of the reviewers found the paper of considerable interest, though we all had concerns about various aspects, several of those concerns being raised by more than one reviewer.

The authors have been variously responsive to the issues i raised (similarly for those raised by other reviewers). Minor issues have been satisfactorily corrected, and relevant sections in methods (which did answer my concerns, but which i had missed in my earlier review) have been highlighted.

However, the authors have pushed back at revising the paper in respect of

- * providing additional "genomic" analyses (my review)

- * more focused narrative (reviewer 2 and 3)

so i think the concerns that I and the other reviewers raised on those matters of overall style and content remain.

One specific comment re line 113: is the ABO/T2D association really "well-established"? A candidate variant paper is cited (ref 15). Please confirm and cite relevant genome-wide association evidence for the association before describing it as "well established".

Reviewer #2 (Remarks to the Author)

Concerns addressed.

Reviewer #3 (Remarks to the Author)

As noted in the prior review, Surhe et al have executed GWAS of the circulating proteome, as measured by an emerging aptamer-based platform. The study was performed in the KORA cohort with validation in the QMDiab cohort. The data from the study will be made freely available online, and will be valuable not only in pathway analyses, but also in the evaluation and further development of aptamer-based proteomic study. The authors provide several hypothesis-generating vignettes connecting protein expression to previously characterized genetic risk variants.

The authors have provided clarifications (and "softening") related to our original comments. In particular, the discussion around the "causality" of trans-acting pQTLs -- and, in particular, the ABO-INSR and FUT3-CA19-9 -- clarifies this point. We also appreciate the clarification/correction of Supplemental Table 2 to clarify that the replication with ELISA assays involved cis-acting pQTLs.

While the manuscript is still very difficult to digest, the strengths are there.

Reviewer #1 (Remarks to the Author):

This is a revision of a manuscript that I saw in its original version at this journal. I can see that all three of the reviewers found the paper of considerable interest, though we all had concerns about various aspects, several of those concerns being raised by more than one reviewer.

The authors have been variously responsive to the issues i raised (similarly for those raised by other reviewers). Minor issues have been satisfactorily corrected, and relevant sections in methods (which did answer my concerns, but which i had missed in my earlier review) have been highlighted.

However, the authors have pushed back at revising the paper in respect of

- * providing additional "genomic" analyses (my review)

- * more focused narrative (reviewer 2 and 3)

so i think the concerns that I and the other reviewers raised on those matters of overall style and content remain.

Response: Thank you very much for your review of the revised manuscript. Please refer to the marked up version of the manuscript for modifications that we applied to increase the clarity of the paper.

One specific comment re line 113: is the ABO/T2D association really "well-established"? A candidate variant paper is cited (ref 15). Please confirm and cite relevant genome-wide association evidence for the association before describing it as "well established".

Response: We confirm that this (blood-type) association has been reported in several studies (incl. ref 15), but not in a GWAS. We believe that the association is robust, but removed the qualifier "well established" in order not to overstate the experimental support.

Reviewer #2 (Remarks to the Author):

Concerns addressed.

Response: Thank you very much for your review of the revised manuscript.

Reviewer #3 (Remarks to the Author):

As noted in the prior review, Surhe et al have executed GWAS of the circulating proteome, as measured by an emerging aptamer-based platform. The study was performed in the KORA cohort with validation in the QMDiab cohort. The data from the study will be made freely available online, and will be valuable not only in pathway analyses, but also in the evaluation and further development of aptamer-based proteomic study. The authors provide several hypothesis-generating vignettes connecting protein expression to previously characterized genetic risk variants.

The authors have provided clarifications (and “softening”) related to our original comments. In particular, the discussion around the "causality" of trans-acting pQTLs -- and, in particular, the ABO-INSR and FUT3-CA19-9 -- clarifies this point. We also appreciate the clarification/correction of Supplemental Table 2 to clarify that the replication with ELISA assays involved cis-acting pQTLs.

While the manuscript is still very difficult to digest, the strengths are there.

Response: Thank you very much for your review of the revised manuscript. Please refer to the marked up version of the manuscript for modifications that we applied to increase digestibility of the manuscript.